# Immunity against *Moraxella catarrhalis* requires guanylate-binding proteins and caspase-11-NLRP3 inflammasomes

Daniel Enosi Tuipulotu[1,†] [ID], Shouya Feng[1,†], Abhimanu Pandey[1] [ID], Anyang Zhao[1], Chinh Ngo[1], Anukriti Mathur[1], Jiwon Lee[2] [ID], Cheng Shen[1] [ID], Daniel Fox[1] [ID], Yansong Xue[1], Callum Kay[1] [ID], Max Kirkby[1], Jordan Lo Pilato[1], Nadeem O Kaakoush[3] [ID], Daryl Webb[2] [ID], Melanie Rug[2] [ID], Avril AB Robertson[4], Melkamu B Tessema[5] [ID], Stanley Pang[6,7], Daniel Degrandi[8], Klaus Pfeffer[8], Daria Augustyniak[9] [ID], Antje Blumenthal[10] [ID], Lisa A Miosge[1], Anne Brüstle[1], Masahiro Yamamoto[11,12], Patrick C Reading[5,13], Gaetan Burgio[1] [ID] & Si Ming Man[1,*] [ID]

## Abstract

*Moraxella catarrhalis* is an important human respiratory pathogen and a major causative agent of otitis media and chronic obstructive pulmonary disease. Toll-like receptors contribute to, but cannot fully account for, the complexity of the immune response seen in *M. catarrhalis* infection. Using primary mouse bone marrow-derived macrophages to examine the host response to *M. catarrhalis* infection, our global transcriptomic and targeted cytokine analyses revealed activation of immune signalling pathways by both membrane-bound and cytosolic pattern-recognition receptors. We show that *M. catarrhalis* and its outer membrane vesicles or lipooligosaccharide (LOS) can activate the cytosolic innate immune sensor caspase-4/11, gasdermin-D-dependent pyroptosis, and the NLRP3 inflammasome in human and mouse macrophages. This pathway is initiated by type I interferon signalling and guanylate-binding proteins (GBPs). We also show that inflammasomes and GBPs, particularly GBP2, are required for the host defence against *M. catarrhalis* in mice. Overall, our results reveal an essential role for the interferon-inflammasome axis in cytosolic recognition and immunity against *M. catarrhalis*, providing new molecular targets that may be used to mitigate pathological inflammation triggered by this pathogen.

**Keywords** cGAS; IL-1; MCC950; STING; TLR4

**Subject Categories** Immunology; Microbiology, Virology & Host Pathogen Interaction

The EMBO Journal (2023) 42: e112558

## Introduction

*Moraxella catarrhalis* is a Gram-negative human respiratory pathogen and a leading cause of otitis media and exacerbation of chronic obstructive pulmonary disease (Murphy & Parameswaran, 2009). Otitis media is one of the most common childhood infections worldwide, with 80% of all children affected by otitis media before school age (Teele *et al*, 1980, 1989). *M. catarrhalis* is amongst the three most common bacterial causes of otitis media, along with *Streptococcus pneumoniae* and non-typeable *Haemophilus influenzae* (Schilder *et al*, 2016). Chronic otitis media can

1   Division of Immunology and Infectious Disease, The John Curtin School of Medical Research, The Australian National University, Canberra, ACT, Australia
2   Centre for Advanced Microscopy, The Australian National University, Canberra, ACT, Australia
3   School of Medical Sciences, UNSW Sydney, Sydney, NSW, Australia
4   School of Chemistry and Molecular Biosciences, The University of Queensland, Brisbane, QLD, Australia
5   Department of Microbiology and Immunology, The University of Melbourne, The Peter Doherty Institute for Infection and Immunity, Melbourne, VIC, Australia
6   Antimicrobial Resistance and Infectious Diseases (AMRID) Research Laboratory, Murdoch University, Murdoch, WA, Australia
7   Department of Microbiology, PathWest Laboratory Medicine-WA, Fiona Stanley Hospital, Murdoch, WA, Australia
8   Institute of Medical Microbiology and Hospital Hygiene, Heinrich-Heine-University Düsseldorf, Düsseldorf, Germany
9   Department of Pathogen Biology and Immunology, Faculty of Biological Sciences, University of Wroclaw, Wroclaw, Poland
10  Frazer Institute, The University of Queensland, QLD, Brisbane, Australia
11  Department of Immunoparasitology, Research Institute for Microbial Diseases, Osaka University, Osaka, Japan
12  Laboratory of Immunoparasitology, WPI Immunology Frontier Research Center, Osaka University, Osaka, Japan
13  WHO Collaborating Centre for Reference and Research on Influenza, Victorian Infectious Diseases Reference Laboratory, The Peter Doherty Institute for Infection and Immunity, Melbourne, VIC, Australia
    *Corresponding author. Tel: +612 61256793; E-mail: siming.man@anu.edu.au
    †These authors contributed equally to this work

lead to clinical sequalae, such as meningitis, facial paralysis and hearing loss (Bluestone, 2000; Schilder *et al*, 2016), which cause substantial health and economic burden globally (Klein, 2000). Chronic obstructive pulmonary disease is the third most common cause of death in the world, resulting in more than 3 million deaths annually (Collaborators, 2017). *M. catarrhalis* infection accounts for 10% of acute exacerbations of chronic obstructive pulmonary disease and is the second most common causative bacterial pathogen after non-typeable *H. influenzae* (Murphy *et al*, 2005; Murphy & Parameswaran, 2009; Wilkinson *et al*, 2017).

A key feature of otitis media and chronic obstructive pulmonary disease are cycles of respiratory inflammation that result in tissue damage and the development of chronic disease. *M. catarrhalis* infection triggers an immune response via membrane-bound Toll-like receptors (TLR) 2, TLR4 and TLR9, leading to the secretion of the pro-inflammatory cytokines TNF and IL-6 (Slevogt *et al*, 2007; Schaar *et al*, 2011; Hassan *et al*, 2012). However, the activities of TLRs are unlikely to fully account for the complexity of the immune response seen in *M. catarrhalis* infection, given that virulence factors of *M. catarrhalis*, such as the ubiquitous surface protein A, inhibit TLR-NF-κB-dependent inflammatory responses (Slevogt *et al*, 2008), and that mice lacking TLR4 can still mount an immune defence to control *M. catarrhalis* infection (Hassan *et al*, 2012). Further, candidate vaccines containing surface proteins from *M. catarrhalis* lack efficacy in reducing the exacerbations of chronic obstructive pulmonary disease (Andreas *et al*, 2022). A better understanding of the host-pathogen interactions during *M. catarrhalis* infection would further inform the development of immunotherapies that attenuate the development of these clinical manifestations. Indeed, *M. catarrhalis* can invade mammalian cells (Slevogt *et al*, 2007; Spaniol *et al*, 2008), yet the role of cytosolic immunity against this pathogen is not known.

Here, we performed a global transcriptomic and targeted cytokine analyses of macrophages to reveal the key immune signatures activated in response to *M. catarrhalis* infection. We uncovered immune signalling pathways controlled by both membrane-bound and cytosolic pattern-recognition receptors. We have further shown that delivery of *M. catarrhalis* lipooligosaccharide (LOS) to the host cell cytoplasm by outer membrane vesicles triggered activation of the cytosolic innate immune sensor, caspase-4/11. This cytosolic sensing pathway led to activation of the NLRP3 inflammasome and pyroptosis in human and mouse macrophages. We identified that type I interferon signalling initiated caspase-11 activation, which is dependent on the expression of *M. catarrhalis*-targeting guanylate-binding proteins (GBPs), collectively promoting inflammation and host defence against *M. catarrhalis* in mice. These findings identified a critical role for cytosolic innate immunity in the host defence against *M. catarrhalis* infection.

# Results

## RNA-sequencing and multiplex analysis identified innate immune signatures of *Moraxella catarrhalis* infection

We performed RNA-sequencing and multiplex biomarker assays on wild-type (WT) primary mouse bone marrow-derived macrophages (BMDMs) to examine the host response to *M. catarrhalis* infection (Fig 1A–C). Gene expression analysis identified 1,365 significantly upregulated genes in WT BMDMs infected with *M. catarrhalis* compared with uninfected WT BMDMs (Fig 1A). Enrichment analysis of these upregulated genes revealed a markedly discrete cluster of related ontology terms including the biological processes, innate immunity, immune sensing, cytokine production, and cellular responses to cytokines (Fig 1B). We then explored changes at the protein level, with 32-plex biomarker assays identifying the expression and release of innate immune cytokines, such as TNF, interleukin-1 (IL)-α, IL-1β, IL-6, KC (also known as C-X-C motif chemokine 1 or CXCL1), IP10 (also known as C-X-C motif chemokine 10 or CXCL10) and RANTES (also known as chemokine ligand 5 or CCL5) (Fig 1C).

The immune signatures of TNF, IL-1, IL-6 and KC suggested a role for membrane-bound Toll-like receptors, whereas IP10 and RANTES further implicated the involvement of cytosolic sensors triggering type I interferon responses, such as cGAS and STING. To investigate the requirement of these innate immune sensors in response to *M. catarrhalis* infection, we infected $Tlr2^{-/-}$, $Tlr4^{-/-}$, $cGAS^{-/-}$ and $Sting^{Gt/Gt}$ (carrying a T596A mutation in *Sting* which abolishes the production of the protein; Sauer *et al*, 2011) BMDMs with *M. catarrhalis* and analysed the release of TNF, IL-6, KC and

---

**Figure 1. *Moraxella catarrhalis* infection activates the non-canonical NLRP3 inflammasome and induces pyroptosis.**

A   Volcano plot profiling of gene expression changes in WT BMDMs infected with *M. catarrhalis* (Ne11, MOI 100) for 8 h relative to untreated WT BMDMs. Upregulated differentially expressed genes (DEGs) are represented by red dots (*P*-value < 0.01, counts per million > 1 and fold-change > 2) within the grey box.
B   Enrichment and similarity analysis of gene ontology terms associated with the significantly upregulated DEGs depicted in (A).
C   Multiplex ELISA of secreted inflammatory cytokines (right) and associated gene expression by RNA-sequencing (left) of WT BMDMs infected with *M. catarrhalis* (Ne11) as in (A).
D   Immunoblot analysis of caspase-1 (Casp-1), caspase-11 (Casp-11), gasdermin D (GSDMD) and GAPDH (loading control) in WT and mutant BMDMs left untreated (Med.) or assessed 20 h after infection with *M. catarrhalis* (Ne11, MOI 100).
E   Release of IL-1β, IL-18, TNF and LDH from BMDMs after treatment as in (D).
F   Brightfield microscopy analysis of WT and mutant BMDMs after treatment as in (D).
G   IncuCyte live-imaging analysis of WT and $Casp11^{-/-}$ BMDM viability after infection with *M. catarrhalis* as in (D), or transfection with LPS.
H   Scanning electron microscopy of WT BMDMs left untreated or assessed after infection as in (D).
I    Transmission electron microscopy of WT BMDMs left untreated or assessed after infection as in (D).

Data information: Arrowheads indicate dead cells (F). Each symbol represents an independent biological replicate (E). ****$P$ < 0.0001 (one-way ANOVA with Dunnett's multiple-comparisons test (E)). Data are from one experiment representative of two (H and I) or three independent experiments (D and F) or pooled from two (G) or three independent experiments (A–C, E; mean and s.e.m. in E and G). Scale bars, 20 μm (F), 2 μm (H and I).
Source data are available online for this figure.

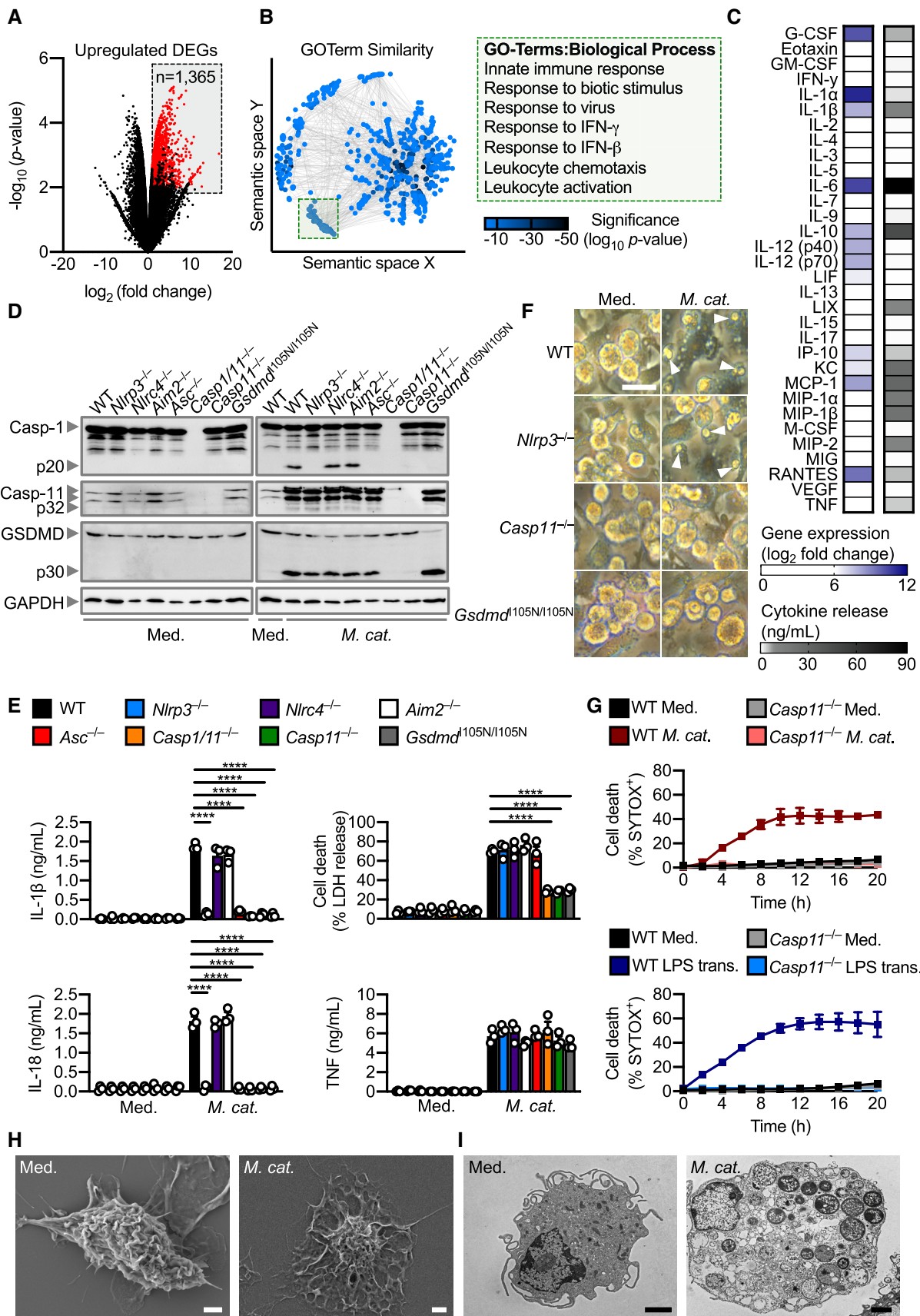

**Figure 1.**

IFN-β. We observed that $Tlr4^{-/-}$, but not $Tlr2^{-/-}$, BMDMs had a substantial reduction in the release of these cytokines, whereas $cGAS^{-/-}$ and $Sting^{Gt/Gt}$ BMDMs exhibited a modest reduction in these responses (Appendix Fig S1A–C). The predominant role of TLR4 is consistent with previous studies showing that TLR4 signalling, in part, contributes to the sensing and clearance of *M. catarrhalis* infection in mice, and that a loss-of-function mutation in TLR4 leads to increased colonisation of *M. catarrhalis* in humans (Hirano *et al*, 2011; Hassan *et al*, 2012; Vuononvirta *et al*, 2013; Terasjarvi *et al*, 2020). TLR4 activity alone, however, cannot completely account for the cytokine profile seen in the global analysis, given that the release of IL-1β requires the biologically-inactive precursor protein called pro-IL-1β to be enzymatically processed by a separate cytosolic immunological pillar called the inflammasome (Bryant & Fitzgerald, 2009; Xue *et al*, 2019).

### *Moraxella catarrhalis* infection activates the Caspase-11-NLRP3 inflammasome

Our global and targeted immunological biomarker analyses revealed the inflammasome-associated cytokine IL-1β, found in acute and chronic otitis media and chronic obstructive pulmonary disease (Lappalainen *et al*, 2005; Botelho *et al*, 2011; Pauwels *et al*, 2011; Damera *et al*, 2016), as one of the highest secreted inflammatory mediators. To investigate this previously unreported role of inflammasomes in *M. catarrhalis* infection, we infected WT BMDMs and BMDMs lacking various inflammasome sensors and components with *M. catarrhalis*. Activation of the inflammasome effector protease, caspase-1, and the release of IL-1β and the related cytokine IL-18 were impaired in $Nlrp3^{-/-}$, $Asc^{-/-}$, $Casp1/11^{-/-}$, $Casp11^{-/-}$, and $Gsdmd^{I105N/I105N}$ (carrying a loss-of-function mutation in the pore-forming protein gasdermin D or GSDMD, which does not impair proteolytic cleavage of GSDMD but prevents pore formation in the plasma membrane; Kayagaki *et al*, 2015) BMDMs infected with *M. catarrhalis*, compared with WT, $Nlrc4^{-/-}$, and $Aim2^{-/-}$ BMDMs (Fig 1D and E). These results suggest a requirement for caspase-11, NLRP3, ASC, caspase-1 and GSDMD in *M. catarrhalis*-induced inflammasome activation.

In the non-canonical inflammasome activation pathway (Kayagaki *et al*, 2011; Broz & Dixit, 2016; Rathinam *et al*, 2019; Pandey *et al*, 2021), caspase-11 is the apical protease activated, which then mediates proteolytic cleavage of GSDMD (Kayagaki *et al*, 2015; Shi *et al*, 2015). Indeed, we observed that the cleavage of caspase-11 and GSDMD, and induction of cell death was impaired in $Casp11^{-/-}$ BMDMs, but not in $Nlrp3^{-/-}$ or $Asc^{-/-}$ BMDMs (Figs 1D–F and EV1A). Live-cell imaging confirmed that cell death was impaired in $Casp11^{-/-}$ BMDMs infected with *M. catarrhalis* compared to WT BMDMs (Fig 1G). Further, pharmacological blockade of NLRP3 using the small-molecule inhibitor MCC950 (Coll *et al*, 2015, 2019; Tapia-Abellan *et al*, 2019) only impaired activation of caspase-1, release of IL-1β and IL-18, but not cleavage of caspase-11 and GSDMD or LDH release (Fig EV1C and D). These results suggest that *M. catarrhalis* infection activates caspase-11 and this is followed by activation of the NLRP3-ASC inflammasome. Importantly, the secreted levels of the pro-inflammatory cytokines TNF, KC, and IL-6 (Figs 1E and EV1E), the gene expression of pro-IL-1β, pro-IL-18, TNF, KC, and IL-6 (Fig EV1F), and the phosphorylation status of IκB and ERK between WT and $Casp11^{-/-}$ BMDMs following infection

with *M. catarrhalis* were similar (Fig EV1B). These data confirmed that the production of inflammasome-independent cytokines and activation of other pro-inflammatory signalling pathways were not affected by the genetic deletion of caspase-11.

Caspase-11-dependent cleavage of GSDMD leads to formation of pores on the plasma membrane, which might mediate the efflux of $K^+$ (Ruhl & Broz, 2015). Inductively Coupled Plasma-Optical Emission Spectrometry (ICP-OES) revealed that the cytosolic concentration of $K^+$, but not $Na^+$, in WT BMDMs infected with *M. catarrhalis* substantially decreased compared to untreated WT BMDMs (Appendix Fig S2A and B). The addition of extracellular $K^+$ to BMDMs infected with *M. catarrhalis* dose-dependently inhibited the activation of caspase-1 and release of IL-1β and IL-18, but not caspase-11 and GSDMD cleavage or LDH release (Appendix Fig S2C and D).

Scanning electron microscopy and transmission electron microscopy revealed that BMDMs infected with *M. catarrhalis* underwent pyroptosis, showing plasma membrane damage, nuclear condensation, structural collapse, and cellular flattening (Fig 1H and I). Our results suggest that *M. catarrhalis* infection triggers activation of caspase-11 which mediates GSDMD cleavage and efflux of $K^+$, leading to pyroptosis and activation of the NLRP3 inflammasome.

### Immune recognition of *M. catarrhalis* by caspase-4/11 is functionally conserved between mice and humans

Effective innate immune sensing of pathogens is, in part, determined by strain-to-strain variations of a pathogen and host-specific differences (Mathur *et al*, 2019; Fox *et al*, 2020; Devant *et al*, 2021; Digby *et al*, 2021). To investigate the possible conservation or divergence in the caspase-11 pathway, we profiled 12 different strains of *M. catarrhalis* for their ability to activate caspase-11 (Table 1). These strains triggered the cleavage of caspase-11, GSDMD and caspase-1, and the release of IL-1β, IL-18 and LDH in WT BMDMs, and substantially less so in $Casp11^{-/-}$ BMDMs (Fig EV2A and B). To verify whether this response was conserved between mice and humans, we infected $Casp4^{-/-}$ THP-1 human cells (lacking caspase-4, the human ortholog of mouse caspase-11) with *M. catarrhalis* and found that these cells had an impaired ability to release IL-1β, IL-18, and LDH, and an impaired ability to undergo cell death compared to WT THP-1 cells (Fig EV2C and D). These findings highlight that innate immune recognition of *M. catarrhalis* by the Caspase-4/11 pathway is functionally conserved between mice and humans.

### Outer membrane vesicles deliver LOS to activate caspase-11

LPS from Gram-negative bacteria can induce oligomerisation and activation of caspase-4 and caspase-11 to induce inflammasome activation (Shi *et al*, 2014). However, *M. catarrhalis* synthesises LOS, which is chemically and structurally different to LPS (Verduin *et al*, 2002; de Vries *et al*, 2009). To investigate whether LOS from *M. catarrhalis* is the ligand which activates the inflammasome, we transfected *M. catarrhalis* LOS and observed cleavage of caspase-1, and the release of IL-1β and IL-18 in WT, but not in $Nlrp3^{-/-}$ or $Casp11^{-/-}$ BMDMs (Fig 2A and B). We also observed that the cleavage of GSDMD and release of LDH was reduced in $Casp11^{-/-}$ BMDMs compared with WT and $Nlrp3^{-/-}$ BMDMs (Fig 2A and B). In response to LOS transfection, BMDMs lacking the DNA sensor

**Table 1.** *Moraxella catarrhalis* isolates used in this study.

| Strain | Isolated from | Source | Reference |
|---|---|---|---|
| Ne11 | Unknown | American Type Culture Collection | Catlin and Cunningham (1961) |
| O35E | Middle ear fluid | Rochester General Hospital Research Institute (Michael Pichichero) | Unhanand et al (1992) |
| ΔlpxA | N/A | Rochester General Hospital Research Institute (Michael Pichichero) | Peng et al (2005) |
| TRIAL-3/16 | Patient sputum | Fiona Stanley Hospital, Western Australia | This study |
| TRIAL-3/165 | Patient sputum | Fiona Stanley Hospital, Western Australia | This study |
| TRIAL-3/22 | ETT sputum | Fiona Stanley Hospital, Western Australia | This study |
| TRIAL-3/133 | Patient sputum | Fiona Stanley Hospital, Western Australia | This study |
| TRIAL-3/179 | ETT sputum | Fiona Stanley Hospital, Western Australia | This study |
| TRIAL-3/180 | Patient sputum | Fiona Stanley Hospital, Western Australia | This study |
| TRIAL-3/173 | Patient sputum | Fiona Stanley Hospital, Western Australia | This study |
| TRIAL-3/208 | Patient sputum | Fiona Stanley Hospital, Western Australia | This study |
| TRIAL-3/218 | Patient sputum | Fiona Stanley Hospital, Western Australia | This study |

ETT, Endotracheal tube.

AIM2 had similar levels of inflammasome activation compared with WT BMDMs, indicating that LOS-induced caspase-11 responses were not interfered by contaminating DNA inducing AIM2 inflammasome activation (Fig 2A and B). To provide unequivocal genetic evidence for the role of LOS in the activation of caspase-11, we used an isogenic mutant strain of *M. catarrhalis* lacking LOS called ΔlpxA, owing to the genetic deletion of the gene encoding the UDP-*N*-acetylglucosamine acyltransferase responsible for the first step of lipid A biosynthesis (Peng *et al*, 2005). We confirmed that genetic disruption of the *lpxA* gene did not affect the size and shape of the bacteria and that the WT and ΔlpxA strains were both efficiently phagocytosed by BMDMs (Fig EV3A–E). We found that WT *M. catarrhalis* induced cleavage of caspase-11, GSDMD and caspase-1, the release of IL-1β, IL-18 and LDH, whereas ΔlpxA *M. catarrhalis* did not (Figs 2C and D, and EV3F). ΔlpxA *M. catarrhalis* also had an impaired ability to trigger the secretion of TNF compared to the isogenic WT strain (Fig 2D). We rescued the priming defect by adding Pam3CSK4 or IFN-γ to BMDMs, and further confirmed that infection of BMDMs with *M. catarrhalis* ΔlpxA indeed did not lead to inflammasome activation (Fig EV3G–I). These findings suggest that LOS of *M. catarrhalis* is the ligand leading to both priming and activation of the inflammasome.

*Moraxella catarrhalis* secretes outer membrane vesicles (OMVs) (Schaar *et al*, 2011; Schwechheimer & Kuehn, 2015; Augustyniak *et al*, 2018), which we hypothesised are key virulence factors which deliver LOS from *M. catarrhalis* into the mammalian cells (Vanaja *et al*, 2016; Santos *et al*, 2018). Indeed, OMVs from two different strains of *M. catarrhalis* induced cleavage of GSDMD and caspase-1, and the release of IL-1β and IL-18 in WT BMDMs, but not in $Casp11^{-/-}$ BMDMs (Fig 2E and F, and Appendix Fig S3A–D). OMV-induced caspase-1 activation and the release of IL-1β and IL-18 were abolished in $Nlrp3^{-/-}$ BMDMs, or blocked by the addition of extracellular $K^+$ or MCC950 in WT BMDMs (Fig 2E and F, and Appendix Fig S3E and F). Further, OMVs from *M. catarrhalis* ΔlpxA did not induce inflammasome activation (Fig 2E and F). We confirmed the absence of LOS in the ΔlpxA strain by silver staining, and that genetic disruption of the *lpxA* gene did not affect bacterial OMV production (Fig 2G–I). Together, these results suggest that *M. catarrhalis* OMVs support the cytosolic cargo delivery of LOS to induce activation of the caspase-11-NLRP3 inflammasome.

**Type I IFN signalling is required for caspase-11 activation**

Our transcriptomic and cytokine analyses revealed a type I IFN signature characterised by the upregulation of many interferon-

**Figure 2.** *Moraxella catarrhalis* infection and secreted OMVs deliver LOS to facilitate inflammasome activation.

A Immunoblot analysis of caspase-1 (Casp-1), caspase-11 (Casp-11) and gasdermin D (GSDMD) in WT, $Nlrp3^{-/-}$, $Casp11^{-/-}$ and $Aim2^{-/-}$ BMDMs left untreated (Med.) or 5 h after transfection with 5 μg of LOS from *M. catarrhalis* (O35E) or 5 μg of LPS from *Escherichia coli*.

B Release of IL-1β, IL-18, TNF and LDH from BMDMs after treatment as in (A).

C Immunoblot analysis of Casp-1, Casp-11 and GSDMD in WT, $Nlrp3^{-/-}$ and $Casp11^{-/-}$ BMDMs left untreated (Med.) or assessed 10 h after infection with *M. catarrhalis* (O35E and ΔlpxA, MOI 100).

D Release of IL-1β, IL-18, TNF and LDH from BMDMs after treatment as in (C).

E Immunoblot analysis of Casp-1, Casp-11 and GSDMD in WT, $Nlrp3^{-/-}$ and $Casp11^{-/-}$ BMDMs left untreated (Med.) or assessed 10 h after incubation with 10 μg of OMVs purified from *M. catarrhalis* (O35E and ΔlpxA).

F Release of IL-1β, IL-18, TNF and LDH from BMDMs after treatment as in (E).

G Silver-stained SDS–PAGE of LOS purified from *M. catarrhalis* (O35E and ΔlpxA).

H, I Transmission electron microscopy of negative stained OMVs purified from *M. catarrhalis* (O35E and ΔlpxA).

Data information: Each symbol represents an independent biological replicate (B, D, F). NS, no statistical significance; **$P < 0.01$; ***$P < 0.001$; ****$P < 0.0001$ (one-way ANOVA with Dunnett's multiple-comparisons test (B, D, F)). Data are from one experiment representative of two (G–I) or three independent experiments (A, C, E) or are pooled from three independent experiments (B, D, F; mean and s.e.m. in B, D, F). Scale bars, 100 nm (H and I).
Source data are available online for this figure.

           

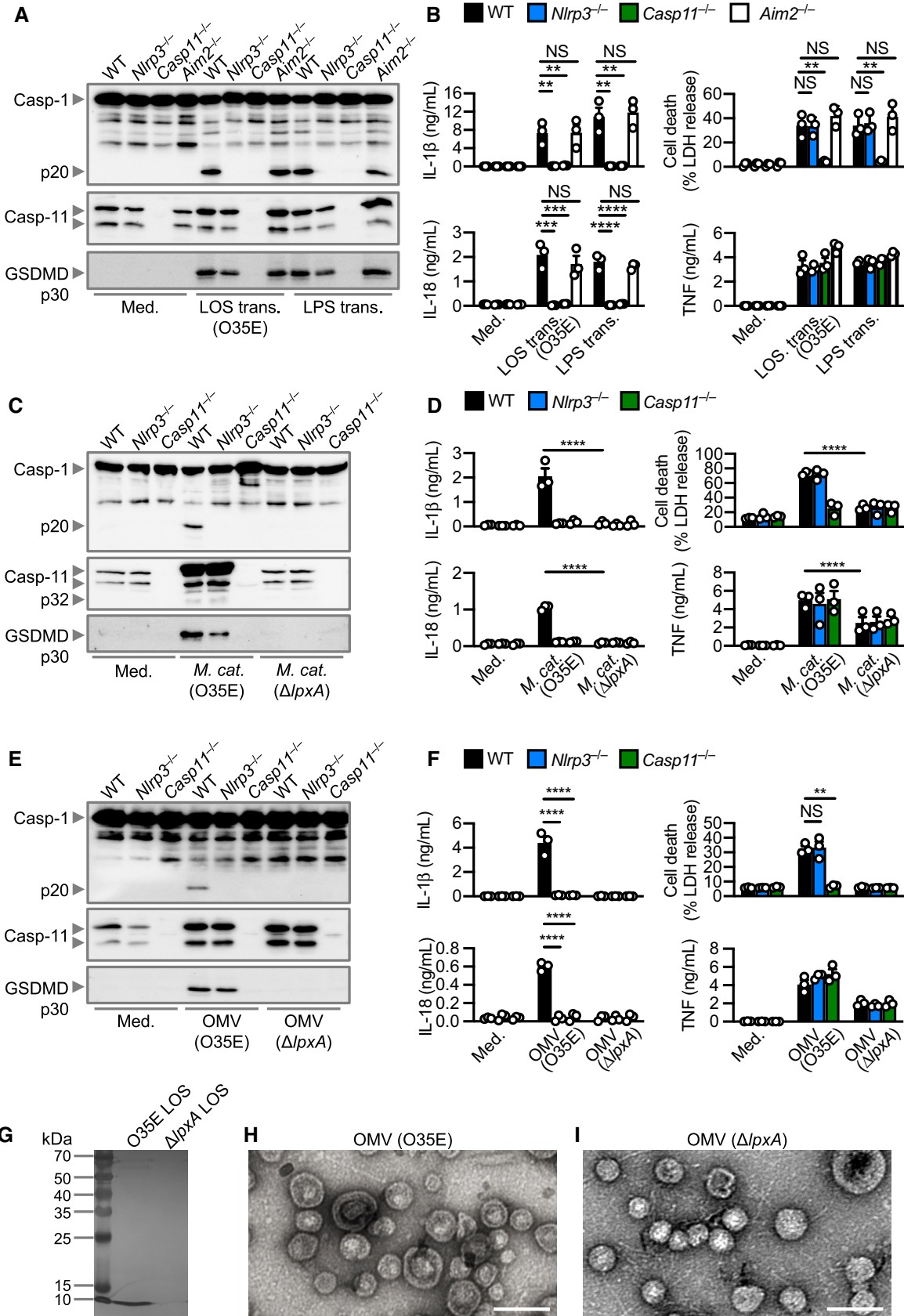

**Figure 2.**

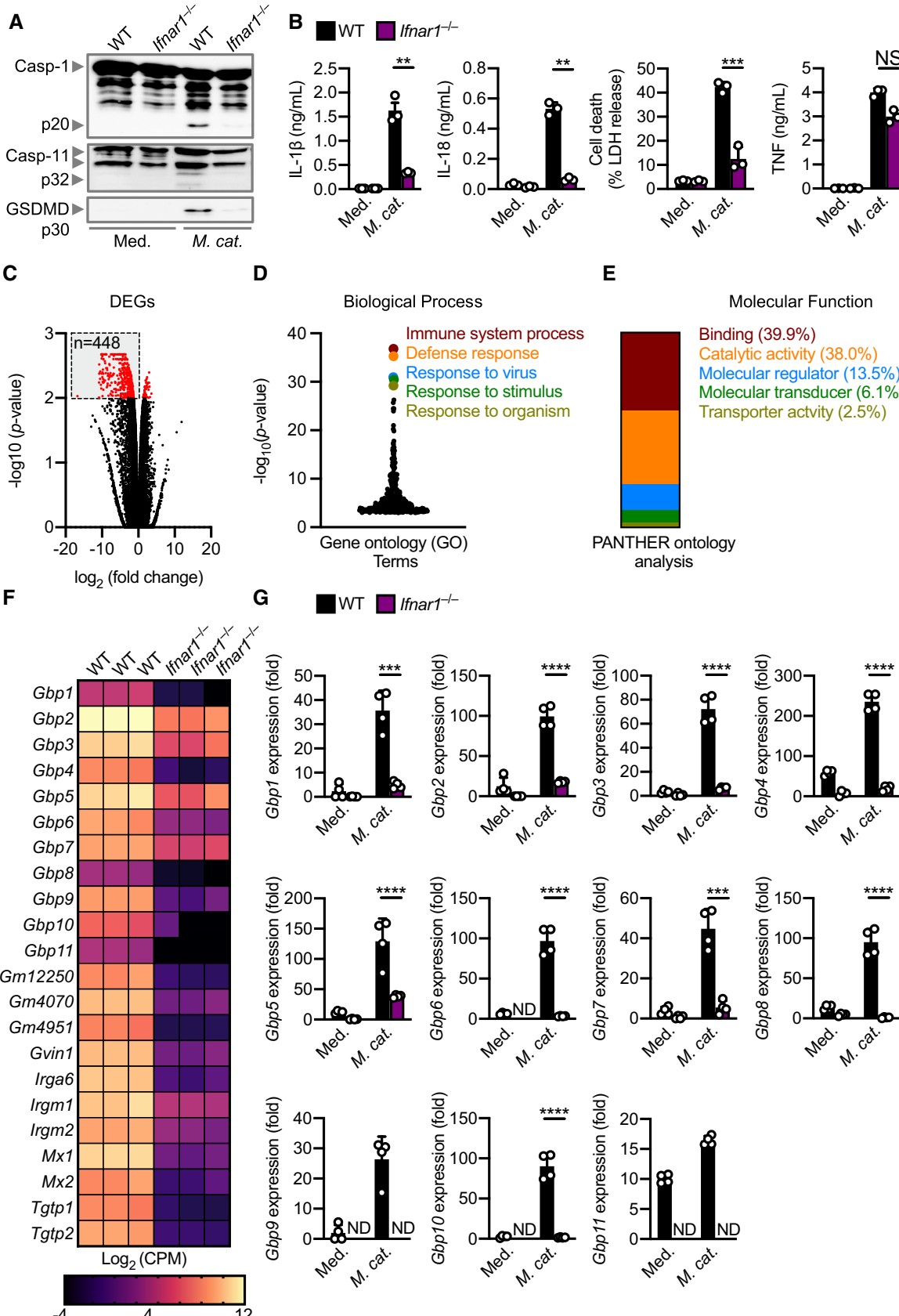

**Figure 3.**

**Figure 3. IFN-inducible proteins are essential for inflammasome activation during *Moraxella catarrhalis* infection.**

A  Immunoblot analysis of caspase-1 (Casp-1), caspase-11 (Casp-11) and gasdermin D (GSDMD) in WT and *Ifnar1*$^{-/-}$ BMDMs left untreated (Med.) or assessed 20 h after infection with *M. catarrhalis* (Ne11, MOI 50).
B  Release of IL-1β, IL-18, LDH and TNF from BMDMs after treatment as in (A).
C  Volcano plot profiling of gene expression changes in *Ifnar1*$^{-/-}$ BMDMs, relative to WT BMDMs, infected with *M. catarrhalis* (Ne11, MOI 100) for 8 h. Differentially expressed genes (DEGs) are represented by red dots. Downregulated DEGs are shown in the grey box.
D  Gene ontology analysis of significantly downregulated DEGs (*n* = 448) as depicted in (C).
E  Functional ontology analysis of the downregulated DEGs present within the top 5 biological processes depicted in (C).
F  Transcript abundance for genes encoding GTPases in WT and *Ifnar1*$^{-/-}$ BMDMs infected with *M. catarrhalis* as in (C). Transcript abundance is provided in counts per million (CPM).
G  qRT–PCR analysis of genes encoding the family of guanylate-binding proteins (GBP1-11) in WT and *Ifnar1*$^{-/-}$ BMDMs left untreated (Med.) or following treatment as in (C).

Data information: Each symbol represents an independent biological replicate (B and G). NS, no statistical significance; **$P$ < 0.01; ***$P$ < 0.001; ****$P$ < 0.0001 (one-way ANOVA with Dunnett's multiple-comparisons test (B and G)). Data are from one experiment representative of three independent experiments (A) or are pooled from three (B–F) or four (G) independent experiments (mean and s.e.m. in B and G). ND, not detected.
Source data are available online for this figure.

stimulated genes such as *Nos2*, *Rsad2*, *Il33*, *Lif*, *Isg15*, *Trim30c*, *Ifit1*, *Oas2*. (Fig 1A–C). How this type I IFN signalling integrates with caspase-11 activation in response to *M. catarrhalis* infection is unknown. Indeed, *Ifnar1*$^{-/-}$ BMDMs lacking the type I IFN receptor and infected with *M. catarrhalis* had an impaired ability to undergo caspase-11, GSDMD, and caspase-1 cleavage, and the secretion of IL-1β, IL-18 and LDH compared to WT BMDMs (Fig 3A and B). To identify the host components connecting type I IFN signalling and inflammasome activation, we performed RNA-sequencing of WT and *Ifnar1*$^{-/-}$ BMDMs left untreated or infected with *M. catarrhalis* (Appendix Fig S4A). We identified 448 genes that were significantly downregulated in *Ifnar1*$^{-/-}$ BMDMs compared to WT BMDMs in response to *M. catarrhalis* infection (Fig 3C). Ontology analysis revealed that these genes were strongly associated with the host response to infection (Fig 3D), with an over representation of genes encoding GTPases carrying binding and catalytic functions (Fig 3D and E). We found that the expression of IFN-inducible GTPases in *Ifnar1*$^{-/-}$ BMDMs infected with *M. catarrhalis* was substantially impaired compared to that in WT BMDMs (Fig 3F), of which, we confirmed defective gene and protein expression of guanylate-

binding proteins (GBPs) in *Ifnar1*$^{-/-}$ BMDMs compared to WT BMDMs (Fig 3G and Appendix Fig S4B).

## GBPs, independently of LOS binding, connect type I IFN signalling and caspase-11 activation

To date, no link between GBPs and *M. catarrhalis* infection has been reported. Immunofluorescence staining revealed that several endogenous GBPs (GBP1, GBP2, GBP3 and GBP5) co-localised with *M. catarrhalis* in BMDMs and mouse lung epithelial cells (Fig 4A–C). These results suggest that GBPs might bind to *M. catarrhalis* to facilitate LOS recognition by caspase-11 during infection. To test this idea, we infected WT BMDMs and BMDMs lacking all five GBPs (GBP1, GBP2, GBP3, GBP5 and GBP7) within the chromosome 3 cluster (called *Gbp*$^{chr3}$-KO) with *M. catarrhalis* (Yamamoto *et al*, 2012; Ngo & Man, 2017). Importantly, we observed that *Gbp*$^{chr3}$-KO BMDMs infected with *M. catarrhalis* had an impaired ability to undergo GSDMD and caspase-1 cleavage, release of IL-1β, IL-18 and LDH compared with WT BMDMs (Fig 4D and E). However, both WT and *Gbp*$^{chr3}$-KO BMDMs transfected with

**Figure 4. GBPs target intracellular *Moraxella catarrhalis*, promote inflammasome activation and mediate bacterial clearance.**

A  Confocal microscopy analysis of intracellular *M. catarrhalis* (green) and GBP1, GBP2, GBP3, GBP5 or GBP7 (red) in WT BMDMs left untreated (Med.) or assessed 12 h after infection with *M. catarrhalis* (Ne11, MOI 20).
B  Quantitation of GBP1-, GBP2-, GBP3-, GBP5- and GBP7-positive *M. catarrhalis* in WT BMDMs as treated in (A).
C  Confocal microscopy analysis of FLAG-OVA, FLAG-GBP1, FLAG-GBP2, FLAG-GBP3, FLAG-GBP5, FLAG-GBP7 (red) and *M. catarrhalis* (green) in LA-4 cells left untreated (Med.), or 16 h after infection with *M. catarrhalis* (Ne11, MOI 20) primed with IFN-γ (100 U/ml).
D  Immunoblot analysis of caspase-1 (Casp-1), caspase-11 (Casp-11) and gasdermin D (GSDMD) in WT, *Gbp*$^{chr3}$-KO and *Casp11*$^{-/-}$ BMDMs left untreated (Med.) or assessed 10 h after infection with *M. catarrhalis* (Ne11, MOI 100), or 5 h after transfection with 5 μg LPS from *Escherichia coli*.
E  Release of IL-1β, IL-18, TNF and LDH from BMDMs after treatment as in (D).
F  Recovery of *M. catarrhalis* (as colony-forming units (CFU)) from primary WT, *Gbp*$^{chr3}$-KO and *Ifnar1*$^{-/-}$ lung fibroblasts 4, 8 and 12 h after infection with *M. catarrhalis* (Ne11, MOI 100).
G  Quantitation of intracellular bacteria by Transmission electron microscopy (TEM) in WT, *Gbp*$^{chr3}$-KO and *Ifnar1*$^{-/-}$ BMDMs infected with *M. catarrhalis* (Ne11, MOI 50) for 12 h.

Data information: Arrowheads indicate bacteria colocalised with GBPs (A and C). Each symbol represents an independent biological replicate (E and F). NS, no statistical significance; *$P$ < 0.05; **$P$ < 0.01; ***$P$ < 0.001; ****$P$ < 0.0001 (one-way ANOVA with Dunnett's multiple-comparisons test (E–G)). To quantify the bacterial number in a field of view, we used the single green channel (anti-*M. catarrhalis* staining), where a single bacterium is defined as an intact circle. To quantify the proportion of GBP-positive bacteria in a field of view, we used the single red channel (anti-GBP staining) and compared this to the green channel (anti-*M. catarrhalis* staining). The number of GBP-positive *M. catarrhalis* was divided by the total number of *M. catarrhalis* counted to obtain the proportion of GBP-positive bacteria (B). A total of 100 BMDMs were analysed to quantify the proportion of GBP-positive bacteria per cell using confocal microscopy (B) or the intracellular bacterial burden using electron microscopy (G). Data are from one experiment representative of two (G) or three independent experiments (A, C, D) or are pooled from two (B) or three independent experiments (E, F; mean and s.e.m. in B, E–G). Scale bars, 5 μm (A, C).
Source data are available online for this figure.

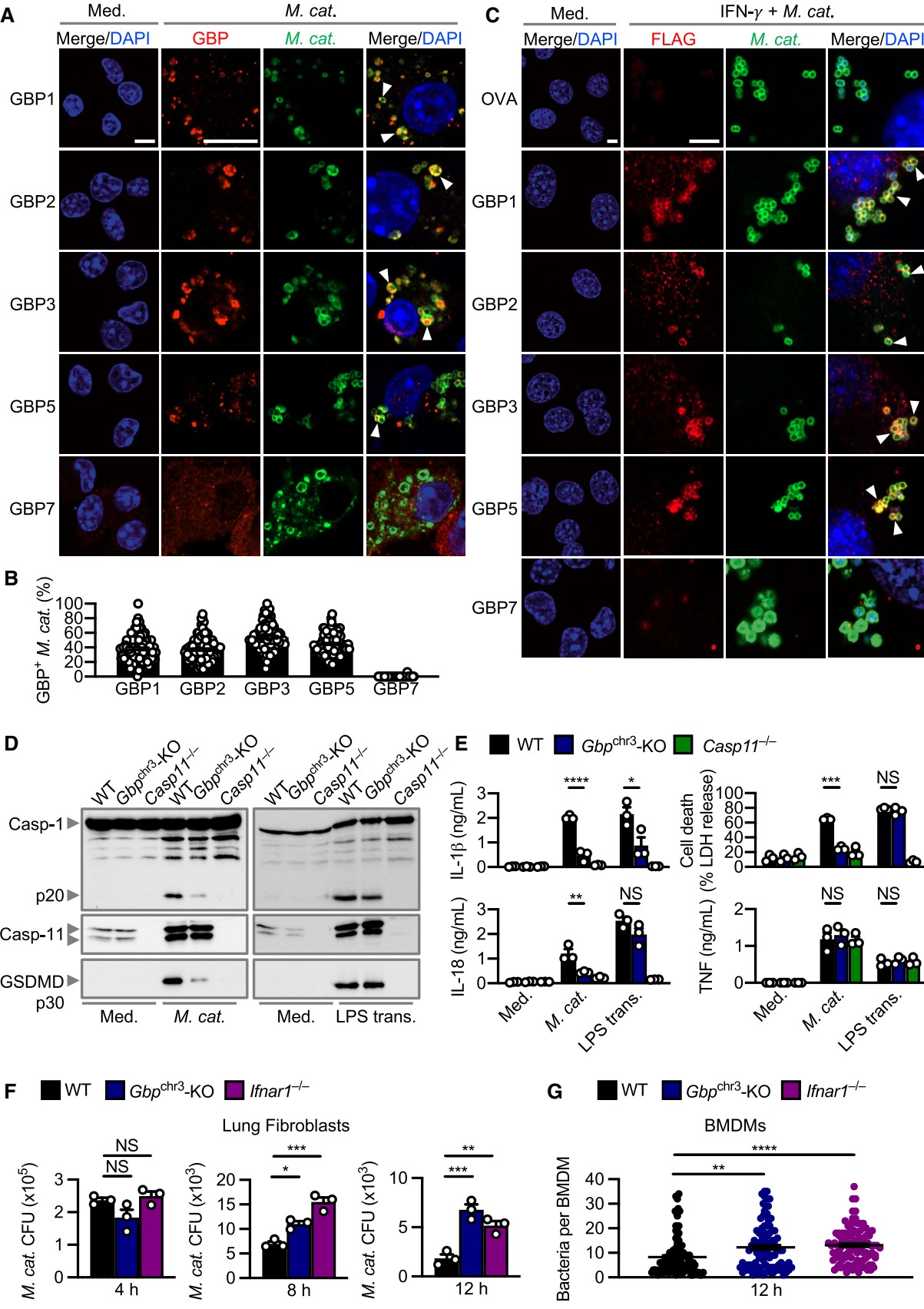

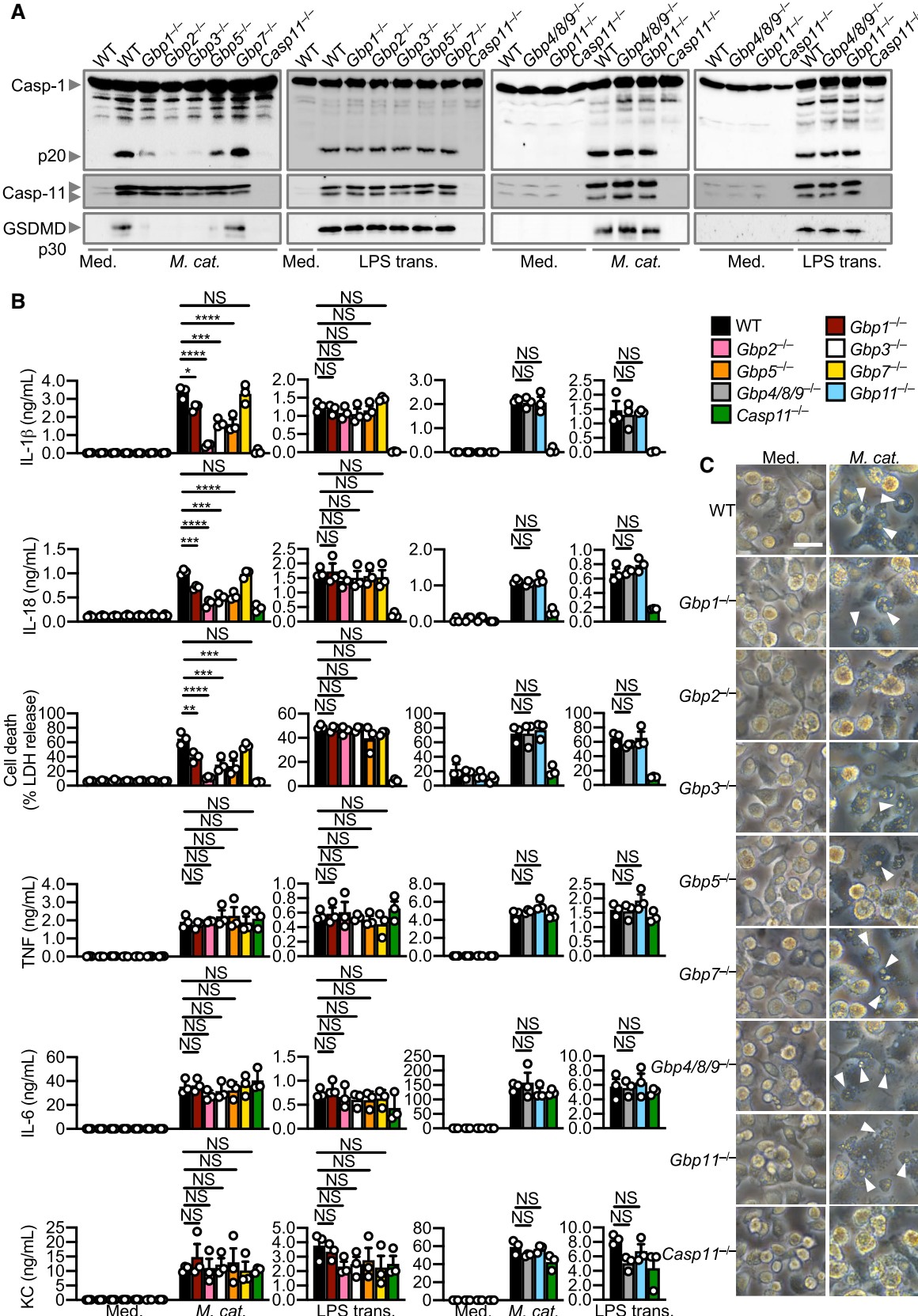

**Figure 5.**

**Figure 5.  GBP1, 2, 3 and 5 are required while GBP4, 7, 8, 9 and 11 are dispensable for *Moraxella catarrhalis*-induced inflammasome activation.**

A   Immunoblot analysis of caspase-1 (Casp-1), caspase-11 (Casp-11) and gasdermin D (GSDMD) in WT, $Gbp1^{-/-}$, $Gbp2^{-/-}$, $Gbp3^{-/-}$, $Gbp5^{-/-}$, $Gbp7^{-/-}$, $Gbp4/8/9^{-/-}$, $Gbp11^{-/-}$ and $Casp11^{-/-}$ BMDMs left untreated (Med.) or assessed 10 h after infection with *M. catarrhalis* (Ne11, MOI 100), or 5 h after transfection with 5 µg LPS from *Escherichia coli*.

B   Release of IL-1β, IL-18, LDH, TNF, IL-6 and KC from BMDMs after treatment as in (A).

C   Brightfield microscopy analysis of WT and mutant BMDMs after treatment as in (A).

Data information: Arrowheads indicate dead cells (C). Each symbol represents an independent biological replicate (B). NS, no statistical significance; *$P < 0.05$; **$P < 0.01$; ***$P < 0.001$; ****$P < 0.0001$ (one-way ANOVA with Dunnett's multiple-comparisons test (B)). Data are from one experiment, representative of three independent experiments (A and C) or are pooled from three independent experiments (B; mean and s.e.m. in B). Scale bars, 20 µm (C).

Source data are available online for this figure.

*M. catarrhalis* LOS undergo GSDMD and caspase-1 cleavage, and the release of IL-1β, IL-18 and LDH (Fig EV4A and B). These data suggest that GBPs encoded on the chromosome 3 cluster are unlikely to sense LOS when it is directly delivered into the cytoplasm. This finding differs to that of human GBP1, which can bind directly to LPS of *Salmonella enterica* serovar Typhimurium and *Shigella flexneri*, and that of mouse GBP5, which can target cytosolic transfected FITC-LPS (Santos *et al*, 2018, 2020; Wandel *et al*, 2020). Instead, we observed colocalisation of mouse GBP1, GBP2, GBP3 and GBP5 with both WT *M. catarrhalis* and the isogenic mutant Δ*lpxA* lacking LOS (Fig EV4C and D). These findings suggest that, unlike LPS (Santos *et al*, 2018, 2020, Wandel *et al*, 2020), LOS does not directly mediate the recruitment of mouse GBPs to the bacteria. The ability of GBPs to target *M. catarrhalis* irrespective of the presence of LOS raised the possibility that specific GBPs might induce bacterial rupture instead of binding to LOS, thereby, liberating LOS from the bacterial cell membrane for sensing by caspase-11. Indeed, $Gbp^{chr3}$-KO and $Ifnar1^{-/-}$ lung fibroblasts or BMDMs infected with *M. catarrhalis* harboured an increased number of intracellular bacteria compared to their WT counterparts (Fig 4F and G).

To narrow down the specific GBPs that are driving inflammasome activation, we used CRISPR-Cas9 technology to generate mouse strains lacking each of the five GBPs on the chromosome 3 locus ($Gbp1^{-/-}$, $Gbp2^{-/-}$, $Gbp3^{-/-}$, $Gbp5^{-/-}$ and $Gbp7^{-/-}$) (Feng *et al*, 2022) and four GBPs on the chromosome 5 locus ($Gbp4/8/9^{-/-}$ and $Gbp11^{-/-}$) (Appendix Fig S5A and B). We observed that $Gbp1^{-/-}$, $Gbp2^{-/-}$, $Gbp3^{-/-}$ and $Gbp5^{-/-}$ BMDMs infected with *M. catarrhalis* had an impaired ability to undergo GSDMD and

caspase-1 cleavage, secretion of IL-1β and IL-18, and cell death compared to WT BMDMs, but the strongest defect was seen in $Gbp2^{-/-}$ BMDMs (Fig 5A–C). However, $Gbp7^{-/-}$, $Gbp4/8/9^{-/-}$ and $Gbp11^{-/-}$ BMDMs responded similarly to WT BMDMs (Fig 5A–C). The production of TNF, IL-6 and KC was not impaired in BMDMs lacking any of the GBPs (Fig 5B), confirming that GBPs mediate inflammasome activation without affecting other major proinflammatory pathways. We independently generated three additional single GBP-knockout mouse strains ($Gbp2^{-/-}$, $Gbp3^{-/-}$ and $Gbp5^{-/-}$) using CRISPR-Cas9 technology (Appendix Fig S6A–F) and confirmed that GBP2 was indeed the dominant GBP driving activation of the inflammasome in response to *M. catarrhalis* infection (Appendix Fig S6G). Further, $Gbp1^{-/-}$, $Gbp2^{-/-}$, $Gbp3^{-/-}$, $Gbp5^{-/-}$, $Gbp^{chr3}$-KO, and $Ifnar1^{-/-}$ BMDMs stimulated with *M. catarrhalis* OMVs had an impaired ability to undergo inflammasome activation, whereas this response was intact in WT, $Gbp7^{-/-}$, $Gbp4/8/9^{-/-}$ and $Gbp11^{-/-}$ BMDMs (Fig EV5A–F).

Given that GBP2 had the strongest effect, we used Correlative Light and Electron Microscopy (CLEM) to localise GBP2 in HEK293 cells infected with *M. catarrhalis* and expressing eGFP-tagged GBP2. We observed that GBP2 was recruited to *M. catarrhalis* in the cytoplasm, with untargeted *M. catarrhalis* also observed in the cytoplasm and vacuoles (Fig 6A). To investigate whether GBPs have antimicrobial activity, we tested the ability of recombinant GBP2 to kill *M. catarrhalis*. We found that recombinant GBP2 inhibited the growth of *M. catarrhalis*, reduced the level of ATP (a measure of bacterial viability), and killed in a dose-dependent manner (Fig 6B–D). To identify potential mechanisms of antimicrobial activity of GBP2, we stained *M. catarrhalis* with a monomeric carbocyanine

**Figure 6.  GBP2 targets and mediates killing of *Moraxella catarrhalis*.**

A   Correlative light electron microscopy (CLEM) of HEK293T cells overexpressing eGFP-GBP2 left untreated (Med.) or infected with *M. catarrhalis* (Ne11, MOI 5).

B   Quantification of $OD_{600}$ of *M. catarrhalis* in BHI media over 6 h in the presence of solvent control (Sol. Ctrl.), 80 µg/ml recombinant GBP2 or 50 µg/ml kanamycin.

C   Quantification of ATP from *M. catarrhalis* in BHI media over 6 h in the presence of Sol. Ctrl., 30 µg/ml recombinant GBP2 or 50 µg/ml kanamycin.

D   Viability of *M. catarrhalis* (as a percentage of CFU in relation to Sol. Ctrl.) assessed 6 h after incubation with recombinant GBP2 at 3, 30 or 300 µg/ml.

E   Quantitation of $DiOC_2(3)$ stained *M. catarrhalis* with fluorescence emission shift from green to red (as a percentage relative to Sol. Ctrl.) assessed 6 h after incubation with recombinant GBP2 at 50 µg/ml or 5 µM CCCP for 30 min.

F   Flow cytometry plots showing the gating strategy for analysis of fluorescence emission by $DiOC_2(3)$ stained bacteria.

G   Uptake of NPN in *M. catarrhalis* (as a percentage relative to 20 µg/ml polymyxin B treatment) assessed 15 min after incubation with Sol. Ctrl. or recombinant GBP2 at 40 µg/ml.

H   Flow cytometric quantitation of SYTOX-stained *M. catarrhalis* treated with 100 µg/ml of recombinant GBP2 for 6 h.

I   Transmission electron microscopy (TEM) analysis of negative stained *M. catarrhalis* 6 h after treatment with Sol. Ctrl. or 100 µg/ml of recombinant GBP2.

J   Scanning electron microscopy (SEM) analysis of the morphology of *M. catarrhalis* 6 h after treatment as in (I).

Data information: White arrowheads indicate bacteria colocalised with GBP2 (A) or ruptured bacteria (I and J). Yellow arrowheads indicate bacteria without GBP2 colocalisation (A). **$P < 0.01$; ***$P < 0.001$; ****$P < 0.0001$ (one-way ANOVA with Dunnett's multiple-comparisons test (B and D) or two-tailed *t*-test (E, G, H)). Data are from one experiment representative of two (A) or three (B, D, G, I, J) independent experiments or are pooled from two (C) or three (E, H) independent experiments (mean and s.e.m. in B–E, G, H). Scale bars, 1 µm (A), 0.5 µm (I and J). Each independent experiment consists of two (C) or three (B, D, G) technical replicates.

Source data are available online for this figure.

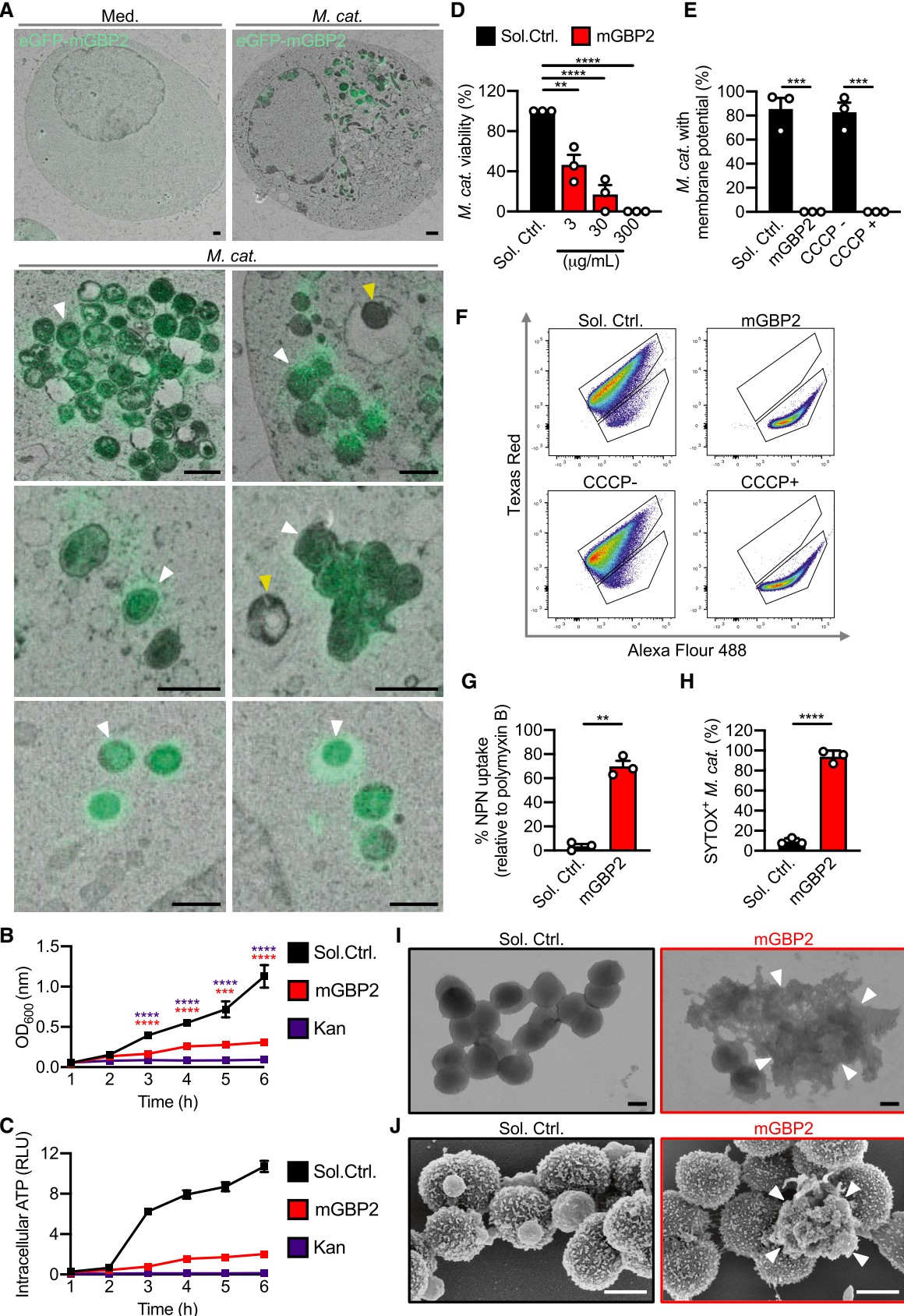

**Figure 6.**

dye DiOC$_2$(3) left untreated or treated with GBP2. In untreated bacterial cells with an intact a membrane potential, DiOC$_2$(3) self-associates to emit a red fluorescence (Novo *et al*, 1999). In *M. catarrhalis* treated with GBP2 or the positive control ionophore CCCP, DiOC$_2$(3) did not aggregate and omitted a green fluorescence, suggesting compromised membrane potential (Fig 6E and F). Further, treatment of *M. catarrhalis* with GBP2 enabled the hydrophobic probe 1-N-phenylnaphthylamine to penetrate the outer membrane of *M. catarrhalis*, and the nucleic acid dye SYTOX to penetrate the inner membrane and stain the DNA of *M. catarrhalis* (Fig 6G and H), indicating that GBP2 disrupts the integrity of both the outer and inner bacterial membrane. Indeed, transmission scanning electron microscopy confirmed that mGBP2 compromised the bacterial membrane integrity, induced membrane rupture and a loss of intracellular content (Fig 6I and J). These findings suggest that GBP2, via its membrane-disruptive activity instead of LOS binding, disrupts the membrane of *M. catarrhalis* to release LOS and facilitate activation of the caspase-11-NLRP3 inflammasome.

## Inflammasomes and GBPs promote host defence against *M. catarrhalis*

We further investigated the role of the Caspase-11-NLRP3 inflammasome in the host defence against *M. catarrhalis* infection *in vivo*. We infected WT, *Casp11*$^{-/-}$ and *Nlrp3*$^{-/-}$ mice intraperitoneally with *M. catarrhalis* and found that the bacterial burden in the spleen was substantially increased in *Casp11*$^{-/-}$ and *Nlrp3*$^{-/-}$ mice compared with WT mice (Fig 7A and C). Analysis of IL-18 in the spleen showed that, compared with WT mice, *Casp11*$^{-/-}$ and *Nlrp3*$^{-/-}$ mice infected with *M. catarrhalis* had an impaired ability to produce this inflammasome-dependent cytokine (Fig 7B and D). Further, *Gbp1*$^{-/-}$, *Gbp2*$^{-/-}$, *Gbp3*$^{-/-}$, and *Gbp5*$^{-/-}$ mice infected with *M. catarrhalis* harboured substantially more bacteria in the spleen compared with WT mice (Fig 7E, G, I and K) and had reduced levels of IL-18 compared with WT mice (Fig 7F, H, J and L). Together, these results highlight a critical role of the inflammasome network in the host defence against *M. catarrhalis* infection.

# Discussion

*Moraxella catarrhalis* infection can lead to otitis media and chronic obstructive pulmonary disease. A recent multicentre and randomised phase 2b trial on a candidate vaccine containing surface proteins from *M. catarrhalis* and the related non-typeable *H. influenza* revealed a lack of efficacy in reducing the yearly rate of moderate or severe exacerbations in patients with chronic obstructive pulmonary disease (Andreas *et al*, 2022). These results indicate that our understanding of the host–pathogen interactions with these bacteria that would necessitate the development of effective vaccines is incomplete. Our finding that caspase-11, on recognition of LOS from *M. catarrhalis*, is essential for host defence revealed another layer of the immune system that is important for controlling *M. catarrhalis* infection. Indeed, detoxified *M. catarrhalis* LOS in combination with one or more *M. catarrhalis* surface proteins have been suggested to be effective vaccine antigens (de Vries *et al*, 2009; Ren & Pichichero, 2016). However, the detoxification process removes most of the lipid A moieties within the LOS, which are the

key components necessitating activation of caspase-4/11. Therefore, vaccine antigens which retain at least some caspase-4/11-activating activity may enhance efficacy.

Our data provide evidence that OMVs can deliver LOS to the host cytoplasm to induce activation of the caspase-11-NLRP3 inflammasome. However, several mechanisms by which LOS is delivered to the host cytoplasm may exist. For example, internalised *M. catarrhalis* likely delivers LOS to the host cytoplasm and extracellular LOS from *M. catarrhalis* may be internalised by CD14 and HMGB1 into the cytoplasm independently of TLR4 internalisation (Deng *et al*, 2018; Vasudevan *et al*, 2022). Given that the lipid A portion of cytoplasmic LPS is recognised by caspase-4/5/11 (Hagar *et al*, 2013; Shi *et al*, 2014), it is likely that lipid A from LOS is also recognised in a similar manner.

Although we provide genetic evidence that *M. catarrhalis* activates caspase-4 in THP-1 cells, our findings could be further strengthened using an NLRP3 and/or caspase-1/11 inhibitor to further interrogate the inflammasome pathway activated by *M. catarrhalis* in human cells. Further, our study did not investigate whether *M. catarrhalis* OMVs or *M. catarrhalis* Δ*lpxA* can induce inflammasome activation in THP-1 cells, and therefore further work is required to characterise the inflammasome signalling pathway in human cells. Other important questions that remain unanswered from our work include whether caspase-5, GBPs and inflammasome signalling contribute to host defence against *M. catarrhalis* in multiple human cell types such as lung epithelial cells.

Our study revealed a mechanistic model whereby TLR4 largely drives the production of interferon and proinflammatory cytokines, including TNF, IL-6 and KC, in response to *M. catarrhalis* infection. Following intracellular invasion, *M. catarrhalis*, presumably its DNA or bacterial cyclic dinucleotides, are sensed by cGAS and STING, which in part contribute to the synthesis of proinflammatory cytokines and IFN-β. We also identified a type I IFN signature defined by GBPs, which provide a signal to trigger caspase-11-NLRP3 inflammasome activation. Of the GBPs investigated, GBP1, 2, 3 and 5 mediated caspase-11 activation, highlighting that this family of proteins may have pathogen-selectivity. Importantly, human GBP1 has been shown to bind to LPS from *S.* Typhimurium or *S. flexneri* (Santos *et al*, 2018, 2020; Wandel *et al*, 2020). Unlike human GBP1, mouse GBPs do not recognise LPS or LOS (Feng & Man, 2020; Feng *et al*, 2022), suggesting that binding between a putative unknown ligand and mouse GBPs may lead to a rupture of the cell membrane of *M. catarrhalis*. This process might liberate LOS into the cytoplasm of mammalian cells for sensing by caspase-11. Indeed, human GBP1 has been shown to disrupt the bacterial O-antigen barrier function, which may also be facilitated by a related IFN-inducible protein called apolipoprotein APOL3 (Kutsch *et al*, 2020; Gaudet *et al*, 2021). However, LOS lacks O-antigen side-chains which are present in LPS (Preston *et al*, 1996), and we observed that GBPs retain their ability to recruit to intracellular *M. catarrhalis* lacking LOS. We reasoned that additional ligands are important for GBP recruitment to *M. catarrhalis*. Further studies are required to identify the ligands of GBPs to establish their role as bona fide pattern-recognition receptors.

Our study suggests that GBPs induce bacterial rupture to facilitate LOS release into the cytoplasm for activation of the inflammasome; however, it is likely that GBP-mediated rupture may also release other bacterial ligands apart from LOS for detection by

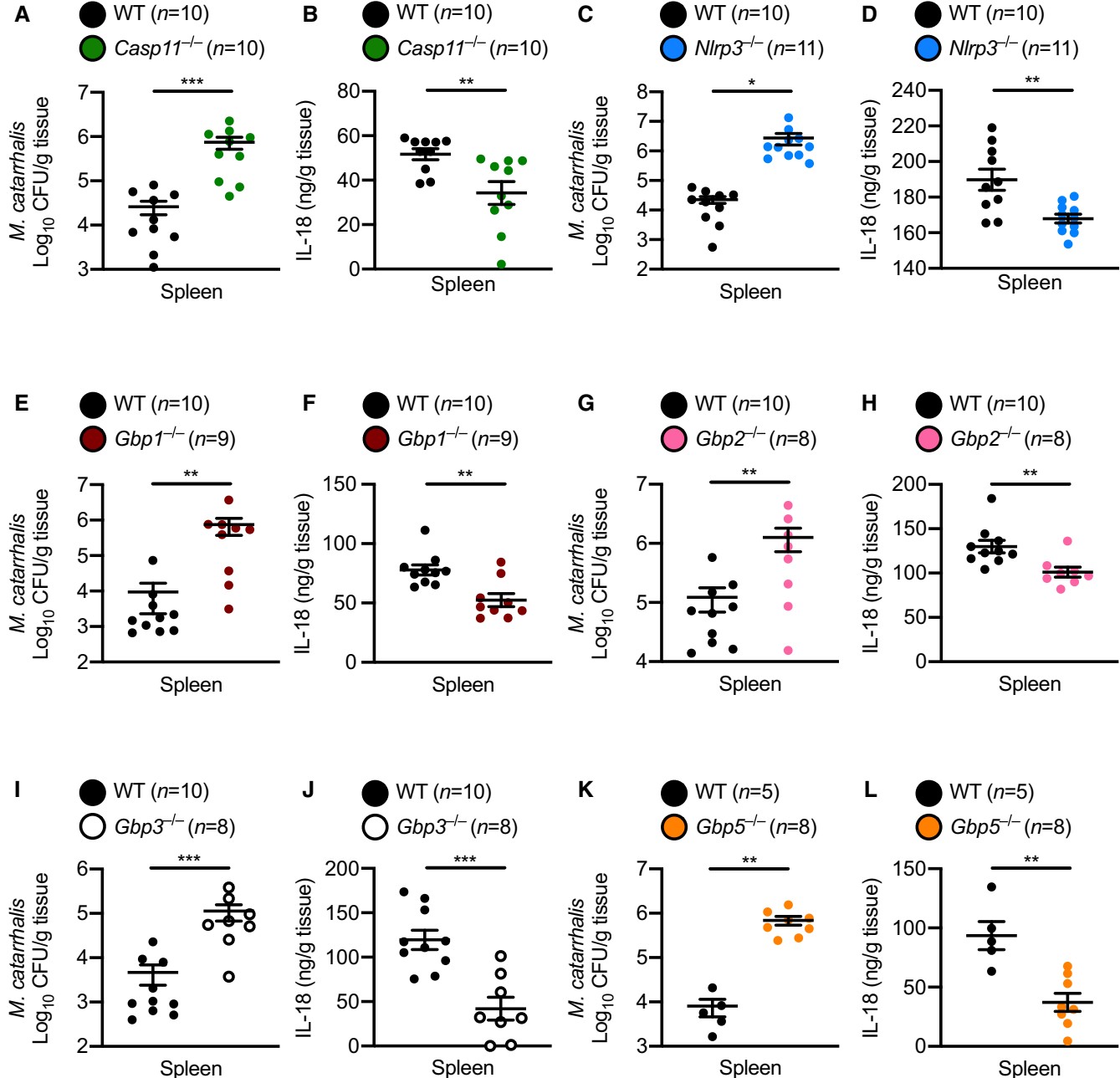

**Figure 7. GBP1, GBP2, GBP3 and GBP5 control bacterial numbers and inflammasome activation during *Moraxella catarrhalis* infection in mice.**

A, B    Bacterial burden and concentration of IL-18 in the spleen of WT mice (*n* = 10) and *Casp11*−/− mice (*n* = 10) after *M. catarrhalis* infection.
C, D    Bacterial burden and concentration of IL-18 in the spleen of WT mice (*n* = 10) and *Nlrp3*−/− mice (*n* = 11) after *M. catarrhalis* infection.
E, F    Bacterial burden and concentration of IL-18 in the spleen of WT mice (*n* = 10) and *Gbp1*−/− mice (*n* = 9) after *M. catarrhalis* infection.
G, H    Bacterial burden and concentration of IL-18 in the spleen of WT mice (*n* = 10) and *Gbp2*−/− mice (*n* = 8) after *M. catarrhalis* infection.
I, J    Bacterial burden and concentration of IL-18 in the spleen of WT mice (*n* = 10) and *Gbp3*−/− mice (*n* = 8) after *M. catarrhalis* infection.
K, L    Bacterial burden and concentration of IL-18 in the spleen of WT mice (*n* = 5) and *Gbp5*−/− mice (*n* = 8) after *M. catarrhalis* infection.

Data information: A mix of male and female mice 6–8 weeks old were used in each experiment and were injected i.p. with $2 \times 10^7$ CFU of *M. catarrhalis* and analysed after 6 h. Each symbol represents an individual mouse (A–L). Each panel represents data from a single experiment (mean and s.e.m. in A–L). Each experiment was performed at least two times. *$P < 0.05$; **$P < 0.01$; ***$P < 0.001$ (two-tailed *t*-test (A–L).
Source data are available online for this figure.

cytosolic immune sensors. Given our finding that cGAS and STING partly contribute to IFN-β production in response to *M. catarrhalis* infection, it is possible that GBPs mediate the release of DNA from *M. catarrhalis* for detection by these sensors to amplify IFN-GBP-inflammasome signalling.

The functional redundancies between GBPs in the activation of inflammasomes have been observed in macrophages infected with *Francisella novicida* (GBP1, GBP2, GBP3, GBP5 and GBP7), *Escherichia coli* (GBP2 and GBP5), and *Citrobacter rodentium* (GBP2 and GBP5) (Man *et al*, 2015, 2016; Finethy *et al*, 2017; Feng *et al*, 2022). In our study, the strongest reduction in inflammasome activation was observed in *Gbp2*$^{-/-}$ BMDMs. However, *Gbp2*$^{-/-}$ BMDMs did not have a complete loss of inflammasome responses, suggesting that other GBPs also in a nuanced way contribute to inflammasome activation in response to *M. catarrhalis*. Indeed, we observed that BMDMs lacking GBP1, GBP3 and GBP5 had a reduction in inflammasome activation in response *M. catarrhalis* infection, highlighting that GBP2 alone is not solely responsible for inflammasome activation. A key question that has remained unanswered is how GBPs individually contribute to the host response. It is possible that each GBP is recruited to *M. catarrhalis* independently of one another, or that GBPs are recruited sequentially, with GBP2 as the apical GBP in the recruitment process. Following the recruitment of GBPs to the bacteria, GBPs may further stabilise one another to enable to the optimal release of PAMPs from bacteria and mediate the activation of inflammasomes.

Although we used an intraperitoneal injection to model the systemic spread of *M. catarrhalis*, the majority of *M. catarrhalis* infections occur within the respiratory tract. Therefore, inhalation and/or intratracheal infection routes could help elucidate the role of inflammasomes and GBPs in the lungs.

Identification of the cytosolic innate immune sensing pathway in the context of *M. catarrhalis* infection may inform therapies targeting overt inflammation leading to exacerbation of diseases such as chronic obstructive pulmonary disease. While a caspase-4 inhibitor is not currently available, several NLRP3 inhibitors are undergoing development and clinical trials for inflammation-associated manifestations, such as gout, atherosclerosis, non-alcoholic steatohepatitis, and COVID-19 (El-Sharkawy *et al*, 2020; Coll *et al*, 2022). Further, GSDMD inhibitors have been described (Hu *et al*, 2020; Humphries *et al*, 2020), of which dimethyl fumarate is already being used in the treatment of multiple sclerosis (Humphries *et al*, 2020). Indeed, administration of inflammasome-specific inhibitors may have short-term benefit of controlling acute pathological inflammation in the lungs of patients with chronic obstructive pulmonary disease, but might also reduce the long-term risk of inflammation-associated lung cancer which are prevalent amongst patients with chronic obstructive pulmonary disease (King, 2015).

## Materials and Methods

### Mice

C57BL/6NCrlAnu mice, *Mefv*$^{-/-}$ (Feng *et al*, 2022), *Gsdmd*$^{I105N/I105N}$ (Kayagaki *et al*, 2015) and *Ifnar1*$^{-/-}$ (Muller *et al*, 1994) mice were obtained from The Australian Phenomics Facility at the Australian National University. *Nlrp3*$^{-/-}$ (Kovarova *et al*, 2012) *Casp1/11*$^{-/-}$

(Kuida *et al*, 1995), *Casp11*$^{-/-}$ (Wang *et al*, 1998), *cGAS*$^{-/-}$ (Schoggins *et al*, 2014) and *Sting*$^{Gt/Gt}$ (Sauer *et al*, 2011) mice were sourced from The Jackson Laboratory. *Nlrc4*$^{-/-}$ (Mariathasan *et al*, 2004), *Asc*$^{-/-}$ (Mariathasan *et al*, 2004), *Tlr2*$^{-/-}$ (Takeuchi *et al*, 1999) and *Tlr4*$^{-/-}$ (Hoshino *et al*, 1999) mice were sourced from the University of Queensland. *Aim2*$^{-/-}$ mice (Jones *et al*, 2010) were sourced from Genentech. *Gbp*$^{chr3}$-KO mice (Yamamoto *et al*, 2012) were sourced from Osaka University. *Gbp1*$^{-/-}$, *Gbp2*$^{-/-}$, *Gbp3*$^{-/-}$, *Gbp5*$^{-/-}$, and *Gbp7*$^{-/-}$ mice were described previously (Feng *et al*, 2022).

The second strain of mice with a genomic deletion of GBP2, GBP3, or GBP5, or mice lacking GBP4/8/9 or GBP11 were generated by Cas9/CRISPR-mediated genome editing technology as previously described (Jiang *et al*, 2019; O'Brien *et al*, 2019). The mouse genomic sequences were obtained from Ensembl (Ensembl.org). Cas9 protein (Cat: 1081059) and the single guide RNA (sgRNA) were purchased from IDT (Singapore) with the following sequences: *Gbp4/8/9* sgRNA1 5'-GGTGGAGGCGGGGTATGGTG-3' and *Gbp4/8/9* sgRNA2 5'-CATGGAGAGTGGAATTTGAG-3' upstream of the *Gbp4* gene and *Gbp4/8/9* sgRNA3 5'-AGCTCACTGCTTCTCCATAC-3' and *Gbp4/8/9* sgRNA4 5'-TTCATCTCTTGTAAGATGGG-3' downstream of *Gbp8* gene. *Gbp11* sgRNA1 5'-ACTGTGCAATCTCAGACCAA-3', *Gbp11* sgRNA2 5'-CACTGAAGCTGAAGTTAAAT-3' and *Gbp11* sgRNA3 5'-TATTCTAGTGACAACCTATGTGG-3' targeting exons 2, 4 and 6, respectively. The gRNA sequences for *Gbp2*, *Gbp3* and *Gbp5* were previously described (Feng *et al*, 2022).

The nucleases were delivered into the pronucleus of C57BL/6NCrlAnu fertilised zygotes at the following concentrations: Cas9 protein (50 ng/μl) was complexed with a sgRNA (2.5 ng/μl) as a ribonucleoprotein complex and micro-injected into the mouse zygotes. After the micro-injection, zygotes were incubated overnight at 37°C under 5% $CO_2$, and two-cell stage embryos were surgically transferred into the uterus of the pseudopregnant CFW/Crl mice. DNA was extracted from the ear punches of the mice using a crude DNA extraction protocol and PCR amplification. PCR products were then purified with a PCR Clean-Up System (Promega) kit according to the manufacturer's instructions. Sanger sequencing was performed in the Biomolecular Resource facilities at the Australian National University and identified an 8,770 bp deletion within *Gbp2*, a 3,406 bp deletion and 6 bp insertion within *Gbp3*, a 3,616 bp deletion and 3 bp insertion within *Gbp5*, a 14,501 bp deletion within *Gbp11* and a 127 kb deletion and 11 bp insertion between the intergenic region upstream of *Gbp4* and the intergenic region downstream of *Gbp8*.

All mice were on the C57BL/6NCrlAnu background or those on a mixed mouse background were backcrossed to the C57BL/6NCrlAnu background for at least 10 generations. Male and female mice of 6–8 weeks old were used. Mice were bred and maintained at The Australian National University under specific pathogen-free conditions. Animal studies in Australia were conducted in accordance with the Protocol Number A2020/19 approved by The Australian National University Animal Experimentation Ethics Committee, in accordance with the National Health and Medical Research Council (NHMRC) code of practice. Animal studies in Poland were approved by the Ethical Review Committee of University of Environmental and Life Sciences of Wroclaw, Poland (no. 70/2013) and complied with the animal experimentation guidelines of the National Institutes of Health guide for the care and use of Laboratory animals (NIH Publications No. 8023, revised 1978).

## Microbial culture

*Moraxella catarrhalis* Ne11 (25238, American Type Culture Collection), *M. catarrhalis* O35E (Rochester General Hospital, New York), and *M. catarrhalis* clinical isolates (Fiona Stanley Hospital, Western Australia) (Table 1) were grown in brain heart infusion (BHI) media (211059, BD) overnight under aerobic conditions at 37°C. Overnight cultures were used directly or subcultured (1:10–1:50) into fresh BHI and grown for 3–4 h to generate log-phase culture. *M. catarrhalis* Δ*lpxA* lacking LOS (Rochester General Hospital, New York) (Table 1) were grown in BHI supplemented with 30 μg/ml kanamycin and cultured as above. Prior to use, Δ*lpxA* bacteria were washed twice with warm PBS to remove residual kanamycin.

## LOS purification

Lipooligosaccharide was isolated from *M. catarrhalis* using the hot phenol-water method described previously with minor modifications (Rezania *et al*, 2011). Briefly, 500 mg of bacterial biomass were resuspended in 10 mM Tris–HCl (pH 8.0) containing 2 mM $MgCl_2$, 2% SDS and 4% mercaptoethanol. Bacteria were treated with proteinase K (100 μg/ml) (19133, Qiagen) at 65°C for 1 h and incubated at 37°C overnight. Two rounds of sodium acetate–ethanol precipitation were performed on the bacterial lysate followed by treatment with DNase I (100 μg/ml) (04716728001, Roche) and RNase A (25 μg/ml) (19101, Qiagen) at 37°C overnight. The crude LOS preparation was heated to 65°C, combined with an equal volume of 90% phenol and incubated at 65°C for 15 min. The mixture was cooled to 4°C, centrifuged at 6,000 *g* for 15 min and the LOS-containing aqueous layer harvested. The purified LOS was dialysed in water to remove residual phenol and concentrated on a 9 kDa spin-filter columns (89884A, ThermoFisher Scientific).

## Purification of OMVs

*Moraxella catarrhalis* OMVs were purified using methods previously described with minor modifications (Vanaja *et al*, 2016). Briefly, 18 h cultures of *M. catarrhalis* (500 ml) were centrifuged at 10,000 *g* for 10 min at 4°C. The supernatant was passed through a 0.22 μm filter and pelleted overnight by ultracentrifugation (150,000 *g*) at 4°C. OMV pellets were resuspended in 200 μl of sterile PBS and streaked onto agar plates to confirm sterility. The total protein content of OMV preparations was quantified by BCA (23225, ThermoFisher Scientific).

Outer membrane vesicles were also generated for mouse immunisation as previously described (Augustyniak *et al*, 2018). Briefly, 18 h cultures of *M. catarrhalis* strains were diluted 50-fold in 500 ml BHI media and incubated at 37°C for 16–18 h, shaking (150 rpm). The cultures were harvested by centrifugation (8,000 *g* for 15 min at 4°C). The supernatants were collected and passed through a 0.22 μm pore size filter vacuum pump (Merck, Millipore). The filtrates were concentrated using 50 kDa Vivaspin centrifugal concentrators (Amicon ultra, Millipore) at 5,000 *g* for 30 min at 4°C. The concentrated supernatants were subsequently pelleted overnight (100,000 *g*, at 4°C) in an ultracentrifuge (Beckman Coulter Optima L-90K, USA). The pellets containing OMVs were re-suspended in 500 μl of sterile PBS buffer (pH 7.4), aliquoted and stored at −20°C. The sterility of the OMV preparations was confirmed on BHI agar.

The protein concentrations in OMVs preparations were measured using Qubit fluorimeter or Bradford assay (Sigma-Aldrich) and the quality of OMVs preparations was confirmed using 12% SDS–PAGE.

## Immunisation and preparation of murine antisera

Antisera were obtained as previously described (Augustyniak *et al*, 2018). Briefly, groups of 5–7 BALB/c mice of similar weight at 8–10 weeks old were immunised intraperitoneally with *M. catarrhalis* OMVs in PBS (protein concentration 100 μg/ml) in 0.1 ml doses, three times, at 2-week intervals (on days 0, 14 and 28). Ten days after the last immunisation, mice were terminally bled through the retro-orbital sinus. Sera from each group immunised with a particular immunogen (OMVs Mc6, OMVs Mc8, and OMVs RH4) were pooled, divided, and stored in aliquots at −70°C until use.

## Bone marrow-derived macrophages

Primary BMDMs were cultured for 5–6 days in Dulbecco's Modified Eagle Medium (DMEM) (11995073, Gibco ThermoFisher Scientific) supplemented with 10% foetal bovine serum (FBS; F8192, Sigma), 30% L929-conditioned media and 1% penicillin and streptomycin (10378016, Gibco ThermoFisher Scientific) as previously described (Man et al, 2016; Jing *et al*, 2022). BMDMs were seeded in antibiotic-free media at a concentration of $1 \times 10^6$ cells per well in 12-well plates.

For activation of the canonical NLRP3 inflammasome, BMDMs were primed using 500 ng/ml ultrapure LPS from *E. coli* (ALX-581-014-L002, Enzo Life Sciences) for 3 h and stimulated with 10 μM nigericin (N7143, Sigma) or 5 mM ATP (10127531001, Roche) for 30 min. For activation of the non-canonical NLRP3 inflammasome, BMDMs were infected with *M. catarrhalis* (MOI 50–100, 10–20 h), treated with 10 μg of OMVs purified from *M. catarrhalis* (10–20 h), transfected with LPS from *E. coli* or LOS purified from *M. catarrhalis*. Each transfection reaction consisted of 5 μg of *E. coli* LPS or 5 μg of *M. catarrhalis* LOS mixed with 0.3 μl of Xfect polymer in Xfect reaction buffer (631318, Clontech Laboratories, Inc.). After 10 min, the mixtures were added to BMDMs in Opti-MEM (31985-070, ThermoFisher Scientific) and incubated for 5–10 h. For inhibition studies, either 20 μM of MCC950 (Coll *et al*, 2015), a selective inhibitor of NLRP3, or an increasing concentration of KCl at 10, 25, 50 and 75 mM (P9541, Sigma) were added to BMDMs 30 min before stimulation.

Bone marrow-derived macrophages were infected with *M. catarrhalis* (MOI 50, 4 h) for qRT–PCR analyses of *Il1b*, *Il18*, *Il6*, *Cxcl1*, *Tnf*, *Ifnb* expression; *M. catarrhalis* (MOI 100, 8 h) for RNA-sequencing and qRT–PCR analyses of *Gbp1*, *Gbp2*, *Gbp3*, *Gbp4*, *Gbp5*, *Gbp6*, *Gbp7*, *Gbp8*, *Gbp9*, *Gbp10*, and *Gbp11* expression; *M. catarrhalis* (MOI 50–100, 10–20 h) for ELISA of IL-1β, IL-18, IL-6, KC and TNF; *M. catarrhalis* (MOI 50, 5–60 min) for WB analyses of pIkB, IkB, pERK, ERK expression; *M. catarrhalis* (MOI 50, 0–24 h) for WB analyses of GBP expression; *M. catarrhalis* (MOI 20, 12–16 h) for immunofluorescence staining of GBPs including 1 h of incubation in gentamicin (50 μg/ml) containing media.

## THP-1 cell culture

Wild-type and *Casp4*$^{-/-}$ (thp-nullz and thp-kocasp4z, Invivogen) THP-1 cells were cultured in RPMI-1640 (11875093, ThermoFisher Scientific) supplemented with 10% foetal bovine serum and 1%

penicillin and streptomycin. Cell lines were authenticated by the supplier for mycoplasma contamination. For studies on human macrophage-like cells, THP-1 cells were seeded in PMA-containing media (50 nM) at a concentration of $1 \times 10^6$ cells per well in 12-well plates and incubated for 24 h. Cells were rested in culture media without PMA for 24 h before stimulation. To induce inflammasome activation, THP-1 macrophage-like cells were infected with *M. catarrhalis* (MOI 50, 6 h), transfected with 5 µg of *E. coli* LPS overnight, or primed with Pam3CSK4 (tlrl-pms, InvivoGen) for 3 h before transfection with 5 µg/ml of poly(dA:dT) overnight.

### Colony forming unit (CFU) assay

To quantify viable intracellular *M. catarrhalis,* primary mouse lung fibroblasts were infected with *M. catarrhalis* (MOI 100) for 3, 7 and 11 h followed by 1 h of incubation in media containing 50 µg/ml gentamicin (Gibco). Cells were incubated for an additional 1 h, washed three times with PBS, lysed in water and scraped from plates. The intracellular bacteria were serially diluted in PBS before plating onto BHI agar.

### Lactate dehydrogenase assay

Levels of lactate dehydrogenase released by cells were determined using the CytoTox 96 Non-Radioactive Cytotoxicity Assay according to the manufacturer's instructions (G1780, Promega).

### IncuCyte cytotoxicity analysis

To track cell viability in real time, BMDMs or THP-1 cells were stimulated in presence of the SYTOX Green nuclear stain that penetrates compromised membranes (1 µM; S7020; Life Technologies). Cell death was monitored over time using the IncuCyte Zoom imaging system (Essen Biosciences) and data were collected using IncuCyte v2018B.

### Cytokine analysis

Cytokine concentrations from BMDMs were calculated using a 32-plex ELISA (MPMCYTMAG-70K-PX32, EMD Millipore), a multiplex ELISA for IL-1β, TNF, KC and IL-6 (MCYTOMAG-70K, EMD Millipore), an IFN-β ELISA (MECY2MAG-73K, EMD Millipore) or an IL-18 ELISA (BMS618-3TEN, ThermoFisher Scientific) according to the manufacturers' instructions. Cytokine levels from THP-1 cells were determined using a human IL-1β ELISA (BMS224-2TEN, ThermoFisher Scientific) or a human IL-18 ELISA (BMS267-2TEN, ThermoFisher) as per the manufacturer's instructions. Multiplex cytokine quantification was performed on a MAGPIX (Luminex) analyser and data was collected using xPONENT v4.2.

### Real-time qRT–PCR analysis

RNA was extracted from BMDMs or tissue using TRIzol (15596018, ThermoFisher Scientific). Isolated RNA was converted into cDNA using the High-Capacity cDNA Reverse Transcription Kit (4368814, ThermoFisher Scientific). RT–PCR was performed and analysed on a QuantStudio 12K Flex Real-Time PCR System (Applied BioSystems) PowerUp SYBR Green Mastermix (A25741, ThermoFisher Scientific). Primer sequences can be found in Table 2.

**Table 2.** Primers used in this study for RT–PCR.

| Target | | Sequence (5′-3′) | Reference |
|---|---|---|---|
| Gbp1 | F | ACCTGGAGACTTCACTGGCT | Yamamoto et al (2012) |
| | R | TTTATTCAGCTGGTCCTCCTGTATCC | |
| Gbp2 | F | CTGCACTATGTGACGGAGCTA | Man et al (2015) |
| | R | CGGAATCGTCTACCCCACTC | |
| Gbp3 | F | CCAGAAAACCAACTGGAACGGAA | Yamamoto et al (2012) |
| | R | TCTCCAGACAAGGCACAGTC | |
| Gbp4 | F | CACAAGCTGAGGAATTGCGT | Yamamoto et al (2012) |
| | R | CTTTCCACAAGGGAATCACCA | |
| Gbp5 | F | CTGAACTCAGATTTTGTGCAGGA | Man et al (2015) |
| | R | CATCGACATAAGTCAGCACCAG | |
| Gbp6 | F | AAGACCATGATATGATGCTGA | Yamamoto et al (2012) |
| | R | GAAAATCCATTTAAGAGAGCC | |
| Gbp7 | F | TTGAGGAAATGCCAGAGGACCAGT | Yamamoto et al (2012) |
| | R | GTCTCCACTATTGATAGCATCCACG | |
| Gbp8 | F | CCACAATGAACATCTGTCCGTGAACC | Yamamoto et al (2012) |
| | R | CCAGAGGGAAACCGTGATTCTGTC | |
| Gbp9 | F | CAAAGCTGAAGAAATAAATGC | Yamamoto et al (2012) |
| | R | GCACAAAATGCTTTTTCGATAAG | |
| Gbp10 | F | CTAACCGGAAGTGTTTTGTC | Li et al (2019) |
| | R | CAGAATCCCTAGTTTATTCCC | |
| Gbp11 | F | GAAAGCTGAGGAAATGAGAAGAG | Yamamoto et al (2012) |
| | R | GCCTTTTCAATCAGTAAAGAGG | |
| Casp11 | F | ACGATGTGGTGGTGAAAGAGGAGC | Gurung et al (2012) |
| | R | TGTCTCGGTAGGACAAGTGATGTGG | |
| Il1b | F | GACCTTCCAGGATGAGGACA | Man et al (2016) |
| | R | AGCTCATATGGGTCCGACAG | |
| Il18 | F | GCCTCAAACCTTCCAAATCA | Man et al (2016) |
| | R | TGGATCCATTTCCTCAAAGG | |
| Il6 | F | CAAGAAAGACAAAGCCAGAGTC | Man et al (2016) |
| | R | GAAATTGGGGTAGGAAGGAC | |
| Cxcl1 | F | CAATGAGCTGCGCTGTCAGTG | Man et al (2016) |
| | R | CTTGGGGACACCTTTTAGCATC | |
| Tnf | F | CATCTTCTCAAAATTCGAGTGACAA | Man et al (2016) |
| | R | TGGGAGTAGACAAGGTACAACCC | |
| Ifnb | F | GCCTTTGCCATCCAAGAGATGC | Man et al (2016) |
| | R | ACACTGTCTGCTGGTGGAGTTC | |
| Gapdh | F | CGTCCCGTAGACAAAATGGT | Man et al (2016) |
| | R | TTGATGGCAACAATCTCCAC | |

### RNA-sequencing and analysis

RNA samples were submitted to BGI Australia (QIMR, Queensland) for poly-A enrichment and transcriptome library construction.

Libraries were sequenced on the DNBseq platform to yield > 55,000,000 reads/sample and read quality was assessed using FastQC. Sequencing reads were mapped to the mm10 (UCSC) reference genome using HISAT2 (Kim *et al*, 2015). Read alignments from HISAT2 were assembled into transcripts with StringTie (Pertea *et al*, 2016) and differential expression analysis was performed using EdgeR (Robinson *et al*, 2010). Default parameters were used for all tools. Differentially expressed genes had the following criteria: *P*-value < 0.01, counts per million (CPM) > 1 and fold-change > 2. Ontological analysis was performed using GOrilla (Eden *et al*, 2009), PANTHER (Thomas *et al*, 2003) and REVIGO (Supek *et al*, 2011).

### Immunoblotting

For caspase-1, caspase-11 and GSDMD immunoblotting, BMDMs and supernatant were lysed in lysis buffer and sample loading buffer containing sodium dodecyl sulphate (SDS) and 100 mM dithiothreitol (DTT). For immunoblotting of GBPs, pIkB, IkB, pERK, ERK, and GAPDH, the supernatant was removed and BMDMs were washed once with PBS, followed by lysis in radioimmunoprecipitation buffer and sample loading buffer containing SDS and 100 mM DTT. Proteins were separated on 8–12% polyacrylamide gels. Following electrophoretic transfer of proteins onto polyvinyldifluoride (PVDF) membranes (IPVH00010, Millipore), membranes were blocked in 5% skim milk in tris-buffered saline with Tween-20 (TBST) and incubated overnight with primary antibodies against caspase-1 (1:3,000 dilution, AG-20B-0042, Adipogen), caspase-11 (1:1,000 dilution, ab180673, Abcam), GSDMD (1:3,000 dilution, ab209845, Abcam), GBP1 (1:1,000 dilution; Virreira Winter *et al*, 2011), GBP2 (1:1,000 dilution, CAC07820, Biomatik), GBP3 (1:1,000 dilution, SA0035 RB1060, Biomatik), GBP5 (1:1,000 dilution; Degrandi *et al*, 2007), GBP7 (1:1,000 dilution, SA0039 RB1065, Biomatik), pIkB (1:1,000 dilution, 2859, Cell Signalling Technologies), IkB (1:1,000 dilution, 9242, Cell Signalling Technologies), pERK (1:1,000 dilution, 9101, Cell Signalling Technologies), ERK (1:1,000 dilution, 9102, Cell Signalling Technologies), GAPDH (1:10,000 dilution, 5174, Cell Signalling Technologies), β-actin (1:10,000 dilution, 8457, Cell Signalling Technologies). PVDF membranes were then incubated with anti-rabbit (1:5,000 dilution, 111035144, Jackson ImmunoResearch) or anti-mouse (1:5,000 dilution, 115035146, Jackson ImmunoResearch) horseradish peroxidase-conjugated secondary antibodies (1:5,000) for 1 h and proteins were visualised using Clarity Western ECL Substrate (170-5061, BioRad) and the ChemiDoc Touch Imaging System (BioRad). Immunoblots were analysed using ImageLab Software v6.01.

### Generation of LA-4 cell lines with doxycycline-inducible expression of GBPs

To produce lentiviruses for the generation of cell lines with doxycycline (DOX) inducible expression of GBPs, a two-step cloning strategy was used. mGBP1 (Accession number NM_010259.2) was engineered to express an N-terminal FLAG-tag and cloned into the pTRE-tight plasmid vector, and then sub-cloned into the pFUV1-mCherry lentivirus transfer plasmid (Herold *et al*, 2008) (kindly provided by Prof Marco Herold, The Walter and Eliza Hall Institute of Medical Research, Parkville, Australia). A cell line expressing cytoplasmic

hen egg ovalbumin (cOVA) lacking the sequence for cell surface trafficking was prepared as a control. Lentivirus stocks to be used for subsequent LA-4 cell transduction were generated in 293T cells following transfection of (i) pMDL (gag and pol), (ii) pRSV-REV packaging plasmids, (iii) pMD2G.VSVg envelope plasmid and (iv) pFUV1-mCherry transfer plasmid by standard procedures. Lentivirus transduced LA-4 cells were sorted based on mCherry-positive cells using a FACSAria III instrument (BD Biosciences, New Jersey, USA).

### GBP overexpression in LA-4 cells

Mouse lung epithelial cells, LA-4, expressing FLAG-tagged OVA, mGBP1, mGBP2, mGBP3 or mGBP5 were cultured in Ham's F-12K (Kaighn's) Medium (21127022, Gibco ThermoFisher Scientific) supplemented with 10% foetal bovine serum and 1% penicillin and streptomycin. LA-4 cells were seeded at a concentration of $1 \times 10^5$ cells per well in 12-well plates. To induce the expression of FLAG-tagged GBPs, LA-4 cells were primed with 10 μg/ml of doxycycline hyclate (D9891, Merck) for 48 h. LA-4 cells were then treated with 100 U/ml of mouse IFN-γ (130-105-790, Miltenyi Biotec) for 24 h before they were infected with *M. catarrhalis* (MOI 20, 16 h) followed by 1 h incubation in gentamicin (50 μg/ml). For immunofluorescence staining of GBPs, LA-4 cells were washed three times with PBS before fixed in 4% paraformaldehyde.

### Immunofluorescence staining

For GBP staining, BMDMs were infected with *M. catarrhalis* for the indicated times and washed three times with PBS. Cells were then fixed with 4% paraformaldehyde at room temperature for 15 min, followed by blocking in 10% normal goat serum (005000121, Jackson ImmunoResearch) supplemented with 0.1% saponin (47036, Sigma) for 1 h. Cells were stained using either rabbit anti-GBP1 (1:20 dilution; Virreira Winter *et al*, 2011), rabbit anti-GBP2 (1:500 dilution; Degrandi *et al*, 2007), rabbit anti-GBP3 (1:20 dilution, SA0035 RB1059, Biomatik) rabbit anti-GBP5 (1:200 dilution; Degrandi *et al*, 2007) or rabbit anti-GBP7 (1:200 dilution; Degrandi *et al*, 2007) overnight at 4°C. An anti-rabbit secondary Rhodamine red antibody (1:500 dilution, 111295144, Jackson ImmunoResearch) was used. Cells were counterstained in DAPI mounting medium (Vecta Labs). GBP staining was visualised and imaged using a Zeiss LSM 800 confocal microscope.

*Moraxella catarrhalis* were stained with anti-*M. catarrhalis* primary antibody (1:500 dilution, anti-RH4 antisera) overnight at 4°C. The secondary antibody used was an Alexa Fluor 488 anti-mouse IgG (1:500 dilution, 115545146, Jackson ImmunoResearch). *M. catarrhalis* were visualised and imaged using a Zeiss LSM 800 confocal microscope.

For co-staining with GBPs, BMDMs were simultaneously stained with the anti-*M. catarrhalis* antibody as above and either anti-GBP1, anti-GBP2, anti-GBP3, anti-GBP5 or anti-GBP7. Alternatively, LA-4 cells were stained with the anti-*M. catarrhalis* antibody as above and anti-DYKDDDDK (1:200 dilution, 2368S, Cell Signalling Technology). Cells were washed five times with PBS. An Alexa Fluor 488 anti-mouse IgG (1:500 dilution, 115545146, Jackson ImmunoResearch) and Alexa Fluor 647 anti-rabbit IgG (1:500 dilution, 111605144, Jackson ImmunoResearch) were used to target the *M. catarrhalis* and GBP primary antibodies or FLAG-tag

(DYKDDDDK) antibody, respectively. Cells were counterstained in DAPI mounting medium (H-1200, Vecta Labs). Bacteria and GBPs were visualised and imaged using a Zeiss LSM 800 confocal microscope. All immunofluorescence data were collected and analysed using ZEN v3.2 (Blue edition).

## Inductively coupled plasma-optical emission spectrometry (ICP-OES) analysis

The intracellular concentrations of $Na^+$ and $K^+$ ions were determined by ICP-OES analysis. Briefly, BMDMs were infected *M. catarrhalis* (MOI 100) for 16 h or primed with LPS for 3 h followed by stimulation with ATP for 30 min. Cells were then washed three times with PBS followed by lysis with concentrated nitric acid ($HNO_3$). The cell lysates were analysed for $Na^+$ and $K^+$ ions performed through ICP-OES using a PerkinElmer OPTIMA 7300 ICP Optical Emission Spectrometer (PerkinElmer).

## Scanning electron microscopy

Bone marrow-derived macrophages or mid-logarithmic phase cultures of *M. catarrhalis* incubated either in the presence or absence of recombinant mGBP2 were washed in PBS, fixed with 2.5% glutaraldehyde in PBS for 3 h and further washed with PBS. Cells were fixed in 1% osmium tetroxide in distilled water for 1 h. Samples were subsequently dehydrated in a series of alcohol and subjected to liquid carbon dioxide critical point drying. Samples were then sputter-coated with platinum (3 nm thickness) at 15 mA for 2 min using the EMI TECH K550 Sputter coater and visualised under a Zeiss UltraPlus Field emission scanning electron microscope at 2–5 kV.

## Transmission electron microscopy

Bone marrow-derived macrophages infected with *M. catarrhalis* (MOI 50–100) for 10–20 h were washed with PBS, fixed in 2.5% glutaraldehyde with 4% paraformaldehyde (15710, Electron Microscopy Sciences) for 15 min and further washed with PBS. Cells were fixed in 1% osmium tetroxide in distilled water for 1 h and stained with 2% uranyl acetate (UA) overnight at 4°C. Samples were subsequently dehydrated in a series of alcohol and then embedded in LR white resin (C023, ProSciTech). Samples were polymerised in a 60°C oven overnight. Thin sections were cut at 80 nm, post-section stained with 2% UA and lead citrate on TEM grids, and viewed using a Hitachi HA7100 transmission electron microscope at 100 kV or a Zeiss Crossbeam focused ion beam scanning electron microscopy (*FIB-SEM*) with STEM detector at 30 kV.

For visualisation of *M. catarrhalis* OMVs or *M. catarrhalis* incubated in the presence of recombinant mGBP2, 2 μl of OMV preparation or 5 μl of mGBP2-treated *M. catarrhalis* culture was absorbed onto glow-discharged carbon-coated TEM grids and negatively stained with 2% UA solution. Samples were viewed using a Hitachi HA7100 transmission electron microscope at 100 kV or a Zeiss Crossbeam FIB-SEM with STEM detector at 30 kV.

## Cloning

For recombinant protein expression, the DNA sequence for mGBP2 (CCDS: 17880.1) was synthesised by Genscript and cloned into

pET28a(+)-TEV between NdeI and XhoI restriction sites, thereby creating a 6x-His Tag at the N-terminus of mGBP2. For overexpression studies, the DNA sequence for mGBP2 (CCDS: 17880.1) was synthesised by Genscript and cloned into pCDNA3.1(+)-N-eGFP between KpnI and XhoI restriction sites.

## Correlative light electron microscopy (CLEM)

HEK293 cells overexpressing eGFP-mGBP2 were infected with *M. catarrhalis* (MOI 5) for 3 h. Cell culture media was replaced with gentamicin-containing media (50 μg/ml) and incubated for 16 h. Cells were harvested and washed with PBS, pelleted by centrifugation and high-pressure frozen (Leica EM-ICE). Cells were then freeze substituted and embedded in resin (Leica EM AFS2). Cells were ultra-microtomed at 300 nm thin sections (Leica EM UC7) and then viewed on confocal microscope (ZEISS LSM800) to capture the fluorescence signal within the cells and on scanning electron microscope (Zeiss UltraPlus FE SEM) to capture ultrastructure. SEM images were captured at 2 kV accelerating voltage using the energy selective backscattered (EsB) detector. After SEM images were captured, the correlation of eGFP-mGBP2 to intracellular *M. catarrhalis* was performed via a shuttle-and-find system (ZEISS).

## Recombinant protein expression and purification

The BL21(DE3) *E. coli* strain (C2527H, NEB) was transformed with pET-28a(+)-TEV plasmid containing the sequence for mGBP2 and transformants were selected with 50 μg/ml kanamycin (10106801001, Roche). A single colony was used to inoculate a starter culture of 10 ml $LB_{Kan}$ broth (LB broth + 50 μg/ml kanamycin) which was incubated at 37°C, shaking (180 rpm) overnight. The overnight culture was diluted 1:100 into 800 ml of $LB_{Kan}$ broth and incubated at 37°C, shaking (180 rpm) for 2–3 h until an $OD_{600}$ of 0.7 was obtained. Cultures were cooled to room temperature, expression was induced by adding isopropyl β-D-1-thiogalactopyranoside (0.5 mM; IPTG, Roche) and the incubation continued at 18°C with shaking (180 rpm) overnight. The culture was centrifuged (5,000 *g*, 20 min, 4°C) to pellet the bacteria and stored at −80°C until required. The cell pellet was resuspended in lysis buffer (50 mM $NaH_2PO_4$, 300 mM NaCl, 10 mM imidazole, 5% glycerol (v/v), 5 mM $MgCl_2$, 0.01% Triton X-100, pH 8.0) supplemented with lysozyme (250 μg/ml), Benzonase nuclease (50 U/ml) and protease inhibitor cocktail (11697498001, Roche) and incubated with gentle agitation at 4°C for 1 h. Cells were subsequently disrupted by sonication and centrifuged (18,000 *g*, 30 min, 4°C) to pellet cellular debris. The supernatant was passed through a 0.22 μm filter (SLGP033RS, Merck) and mGBP2 was purified using Ni-NTA agarose resin (30210, Qiagen) as per the manufacturers' instructions. The purity of eluted proteins was analysed by SDS–PAGE and Coomassie blue staining. Purified proteins were dialysed in DPBS (14190, ThermoFisher) containing 20 mM Tris and 20% glycerol (v/v), pH 7.5.

## Antimicrobial assays

Bacterial viability was measured using colony forming unit (CFU) assays. Briefly, overnight cultures of bacteria were washed and resuspended with PBS to a concentration of $1 \times 10^6$ CFU/ml

respectively. Bacteria were then treated with solvent control, recombinant mGBP2 at the desired concentration and incubated at 37°C for 6 h. Treated bacteria were serially diluted, plated onto BHI agar plates and incubated overnight at 37°C.

Bacterial viability was also measured by measuring optical density ($OD_{600}$) or by quantifying levels of ATP. Briefly, an overnight culture of bacteria was diluted 1:150 into fresh BHI. Bacteria were then treated with solvent control, kanamycin or recombinant mGBP2 at the desired concentration and incubated at 37°C for 6 h with shaking at 180 rpm. Samples were taken hourly for $OD_{600}$ measurements or quantitation of ATP using the BacTiter-Glo Microbial Cell Viability Kit (G8230, Promega).

Bacterial outer membrane permeabilisation was investigated using 1-N-phenylnaphthylamine (NPN) dye uptake. Briefly, an overnight culture of bacteria was diluted 1:50 into fresh BHI and incubated at 37°C for 3–4 h until cultures reached an $OD_{600}$ of 0.5. Bacteria were washed and resuspended in assay buffer (5 mM HEPES, 5 mM glucose, pH 7.2) at half the original culture volume to yield an $OD_{600}$ of 1.0. Resuspended bacteria (100 μl) were added to 96-well plates containing, 50 μl NPN dye and 50 μM of either solvent control, recombinant mGBP2 or polymyxin B at the desired concentration. Samples were incubated for 10 min and fluorescence was recorded on TecanPro200 plate reader; $\lambda_{Ex} = 350$ nm and $\lambda_{Em} = 420$ nm. NPN uptake was calculated as $(t)$ (%) $= (F_{t15} - F_{t0}) \times 100/(F_{t100} - F_{t0})$, where the fluorescence from untreated bacteria was defined as $F_{t0}$ and the fluorescence from bacteria treated with polymyxin B was defined as $F_{t100}$.

Bacterial inner membrane permeabilisation was investigated using SYTOX Green incorporation. Briefly, an overnight culture of bacteria was washed and resuspended with PBS to a concentration of $1 \times 10^9$ CFU/ml. Bacteria were then treated with solvent control or recombinant mGBP2 at the desired concentration and incubated at 37°C for 12 h. After washing with PBS, bacteria were stained with SYTOX Green (5 μM; S7020; Life Technologies) followed by washing with PBS and fixing in 4% PFA (20 min, room temperature). The fluorescence intensity for individual bacteria was measured by flow cytometry (BD LSRII cytometer).

Bacterial membrane potential was assessed using the BacLight Membrane Potential kit (B34950, ThermoFisher). Briefly, an overnight culture of bacteria was washed and resuspended in PBS to a concentration of $1 \times 10^9$ CFU/ml. Bacteria were treated with solvent control and recombinant mGBP2 at the desired concentration or with proton ionophore, carbonyl cyanide 3-chlorophenylhydrazone (CCP) as a positive control. Bacteria treated with mGBP2 were incubated at 37°C for 6 h followed by treatment with the fluorescent membrane-potential indicator dye, $DiOC_2(3)$, as per the manufacturer's instructions. The red- and green-fluorescent bacterial populations were differentiated by flow cytometry (BD LSRII cytometer). All flow cytometry data was collected using BD FACSDiva and analysed using FlowJo v10.7.

### Animal infection

*Moraxella catarrhalis* strain Ne11 was grown as described above. To assess bacterial burden, mice were injected intraperitoneally with $2 \times 10^7$ CFU of *M. catarrhalis* in 200 μl PBS. After 6 h, spleens were harvested and homogenised in PBS with metal beads for 2 min using the Qiagen TissueLyser II apparatus. *M. catarrhalis* CFU were determined by plating lysates onto BHI agar and incubated overnight at 37°C. No randomisation or blinding was performed. No statistical methods were used to calculate sample sizes.

### Data collection and statistical analysis

At least three independent biological repeats were performed for each experiment unless otherwise stated in the figure legend. Experiments were performed without technical replicates unless otherwise stated in the figure legend. For example, cells from one mouse were stimulated in one well of a tissue-culture plate and analysed using various techniques. This was then repeated on at least three separate occasions unless otherwise stated in the figure legend. The GraphPad Prism 9.0 software was used for data analysis. Data are shown as mean ± s.e.m. Statistical significance was determined by *t*-tests (two-tailed) for two groups or one-way analysis of variance (with Dunnett's multiple-comparisons test) for three or more groups. $P < 0.05$ was considered statistically significant.

# Data availability

The RNA-sequencing data generated is accessible through the Gene Expression Omnibus GSE196164 (https://www.ncbi.nlm.nih.gov/geo/query/acc.cgi?acc=GSE196164). All other data generated or analysed during this study are included within the paper. All unique biological materials generated in this study are available from the corresponding author.

Expanded View for this article is available online.

## Acknowledgements

We would like to thank Dr. V.M. Dixit (Genentech, USA), Dr. K. Schroder (Institute for Molecular Bioscience, Australia), Dr. E. Frickel (The Francis Crick Institute, UK), Dr. Michael Pichichero (Rochester General Hospital Research Institute, USA), Dr. Seth Masters and Dr. Marco Herold (The Walter and Eliza Hall Institute of Medical Research, Australia) for reagents. We thank Ms. Cathy Gillespie, Dr. Frank Brink, Dr. Hua Chen, Dr. Jenna Lowe, Ms. Nikki Ross, Ms. Huiming Yang, Ms. Jing Gao and Ms. Lora Starrs (ANU Australia) for technical assistance. The authors acknowledge The National Collaborative Research Infrastructure Strategy (NCRIS) via Phenomics Australia. The authors acknowledge the facilities and the scientific and technical assistance of Microscopy Australia at the Centre for Advanced Microscopy (ANU, Australia) a facility that is funded by the University and the Federal Government. DET is supported by the Gastroenterology Society of Australia Mostyn Family Grant. DET and SF are supported by the Gretel and Gordon Bootes Medical Research Foundation. SF is supported by the John Curtin School of Medical Research PhD Scholarship. SMM is supported by the Australian National University and the National Health and Medical Research Council of Australia (under Project Grants APP1141504, APP1146864, APP1162103 and APP1163358) and a CSL Centenary Fellowship. Open access publishing facilitated by Australian National University, as part of the Wiley - Australian National University agreement via the Council of Australian University Librarians.

## Author contributions

**Daniel Enosi Tuipulotu:** Conceptualization; data curation; formal analysis; investigation; methodology; writing – original draft; writing – review and

editing. **Shouya Feng:** Conceptualization; data curation; formal analysis; investigation; methodology; writing – original draft; writing – review and editing. **Abhimanu Pandey:** Formal analysis; investigation; writing – review and editing. **Anyang Zhao:** Formal analysis; investigation; writing – review and editing. **Chinh Ngo:** Formal analysis; investigation; writing – review and editing. **Anukriti Mathur:** Formal analysis; methodology; writing – review and editing. **Jiwon Lee:** Formal analysis; investigation; writing – review and editing. **Cheng Shen:** Investigation; writing – review and editing. **Daniel Fox:** Investigation; writing – review and editing. **Yansong Xue:** Investigation; writing – review and editing. **Callum Kay:** Investigation; writing – review and editing. **Max Kirkby:** Investigation; writing – review and editing. **Jordan Lo Pilato:** Investigation; writing – review and editing. **Nadeem O Kaakoush:** Formal analysis; investigation; writing – review and editing. **Daryl Webb:** Formal analysis; investigation; writing – review and editing. **Melanie Rug:** Formal analysis; writing – review and editing. **Avril AB Robertson:** Resources; writing – review and editing. **Melkamu B Tessema:** Resources; writing – review and editing. **Stanley Pang:** Resources; writing – review and editing. **Daniel Degrandi:** Resources; writing – review and editing. **Klaus Pfeffer:** Resources; writing – review and editing. **Daria Augustyniak:** Resources; writing – review and editing. **Antje Blumenthal:** Resources; writing – review and editing. **Lisa A Miosge:** Resources; writing – review and editing. **Anne Brüstle:** Resources; writing – review and editing. **Masahiro Yamamoto:** Resources; writing – review and editing. **Patrick C Reading:** Resources; writing – review and editing. **Gaetan Burgio:** Resources; methodology; writing – review and editing. **Si Ming Man:** Conceptualization; data curation; supervision; funding acquisition; writing – original draft; project administration; writing – review and editing.

## Disclosure and competing interests statement

AABR declares that they are a named inventor on inflammasome inhibitor patents (WO2018215818, WO2017140778, and WO2016131098). All other authors declare no competing interests.

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
