## [Review Process File · The EMBO Journal]

Immunity against *Moraxella catarrhalis* requires guanylate-binding proteins and caspase-11-NLRP3 inflammasomes

Daniel Enosi Tuipulotu, Shouya Feng, Abhimanu Pandey, Anyang Zhao, Chinh Ngo, Anukriti Mathur, Jiwon Lee, Cheng Shen, Daniel Fox, Yansong Xue, Callum Kay, Max Kirkby, Jordan Lo Pilato, Nadeem Kaakoush, Daryl Webb, Melanie Rug, Avril Robertson, Melkamu Tessema, Stanley Pang, Daniel Degrandi, Klaus Pfeffer, Daria Augustyniak, Antje Blumenthal, Lisa Miosge, Anne Brüstle, Masahiro Yamamoto, Patrick Reading, Gaetan Burgio, and Si Ming Man

DOI: [10.15252/emboj.2022112558](https://doi.org/10.15252/emboj.2022112558)

Corresponding author(s): Si Ming Man (siming.man@anu.edu.au)

Review Timeline:

Submission Date:	8th Sep 22
Editorial Decision:	20th Oct 22
Revision Received:	25th Nov 22
Editorial Decision:	21st Dec 22
Revision Received:	17th Jan 23
Accepted:	19th Jan 23

Editor: Karin Dumstrei

Transaction Report:

Dear Si Ming,

Thank you for submitting your manuscript to The EMBO Journal. Your study has now been seen by two referees and their comments are provided below. A third referee had agreed to review the manuscript for us, but as I haven't heard back from the referee I will move forward with the two reports on hand.

As you can see below, the referees find the analysis interesting and suitable for The EMBO Journal. They raise a number of comments that I would like to ask you to address in a revised version. I think it would be helpful to discuss the raised points further and I am available to do so via email or video.

I thank you for the opportunity to consider your work for publication. I look forward to discuss your revisions further.

with best wishes

Karin

Karin Dumstrei, PhD
Senior Editor
The EMBO Journal

At EMBO Press we ask authors to provide source data for the main and EV figures. Our source data coordinator will contact you to discuss which figure panels we would need source data for and will also provide you with helpful tips on how to upload and organize the files.

I have attached a guide with helpful tips on how to prepare the revised version

Guide For Authors: <https://www.embopress.org/page/journal/14602075/authorguide>

I realize that it is difficult to revise to a specific deadline. In the interest of protecting the conceptual advance provided by the work, we recommend a revision within 3 months (18th Jan 2023). Please discuss the revision progress ahead of this time if you require more time to complete the revisions.

As a matter of policy, competing manuscripts published during this period will not negatively impact on our assessment of the conceptual advance presented by your study.

Use the link below to submit your revision:

Referee #1:

The manuscript "Cytosolic immunity against *Moraxella catarrhalis* requires innate immune signalling and inflammasomes" by Tuipulotu et al. shows that GBPs compromises *M. catarrhalis* membrane integrity leading to the delivery of *M. catarrhalis* lipooligosaccharide (LOS) to the host cytoplasm by outer membrane vesicles (OMVs) which leads to activation of caspase-11 and subsequently to activation of NLRP3 and pyroptosis.

In the first part of the paper they show that CASP11 is upstream of the activation of NLRP3/CASP1, secretion of IL1b/IL18, and cell death. They go on to show that LOS and OMVs phenocopy the results obtained from infection with whole bacteria. They subsequently show that the inflammasome activation is IFNAR and GBPchr3 dependent, GBPs co-localize with the intracellular bacteria, and deletion of GBP2 has the largest effect on inflammasome activation. Recombinant GBP2 can kill bacteria and GBP2 appears to be able to disrupt the bacterial membrane. Mice lacking GBP1, 2, 3, 5, CASP11, or NLRP3 have higher bacterial burdens and lower levels of IL18 in the spleen.

This is a very detailed and quite complete manuscript that uncovers the mechanism by which *M. catarrhalis* induces the activation of innate immunity. Although the experiments were well performed and the conclusions are valid, the role of GBP2 in mediating the release of PAMPs from bacteria leading to CASP11 activation is not novel and has been described in several manuscripts.

Major comments:

It was unclear how the different GBPs that play a role (GBP1, 2, 3, 5) act together to activate the inflammasome. If GBP2 is sufficient to release PAMPs from the bacteria why does knocking out GBP 1, 3 or 5 also lead to a significant decrease in inflammasome activation?

minor comments:

Fig 1C the cytokine release is not indicated as fold-change over uninfected but as a concentration. It is unclear how this would confirm the gene expression data presented.

Referee #3:

Previous studies have shown that caspase-11 in mice and caspase-4/5 in humans are cytosolic innate immune sensors for LPS and recognition of LPS by these sensors leads to the noncanonical inflammasome activation and pyroptosis executed by GSDMD. Pore formation mediated by N-terminal domain of GSDMD allows not only the release of cytokines of the IL-1 family but also endogenous DAMPs from pyroptotic cells. In addition, interferon induced guanylate-binding proteins (GBPs) have been shown to recognize pathogen-containing vacuoles (*Toxoplasma gondii*, *Listeria monocytogenes*, etc.) or bind directly to the surface of pathogen (*Shigella flexneri*) residing in the host cytosol, leading to lysis of this intracellular niche, disruption of bacterial membrane integrity and induction of inflammasome. However, little has been known for the roles of inflammasome and GBPs in host defense against *Moraxella catarrhalis*, a major causative agent of otitis media and chronic obstructive pulmonary disease. Here, Tuipulotu et al present data suggesting that outer membrane vesicles introduced lipooligosaccharide of *M. catarrhalis* into the host cytosol activates the caspase-4/11, gasdermin-D-dependent pyroptosis, followed by the NLRP3 inflammasome activation in both human and mouse macrophages. They further showed the evidences that GBPs, especially GBP2 in mice, are critical for the inflammasome activations in host cells post *M. catarrhalis* infection and GBP2 can lead to the rupture of *M. catarrhalis* cell membrane and even shows killing activity to this bacteria. Finally, using the animal infection experiments, the authors showed the essentiality of inflammasomes and GBPs for the host defense against *M. catarrhalis*. This study overall is well designed and reports previously unrevealed roles of canonical and non-canonical inflammasome as well GBPs in fighting against *M. catarrhalis*, which will provide valuable information on *M. catarrhalis* and the LOS/LPS activated non-canonical inflammation, as well as the roles played by GBPs. There are still some concerns about this study as listed below.

Major points:

1. The title of the manuscript should be more specific and focused on the cytosolic innate immunity against *M. catarrhalis* discussed in this study, since there are a lots of well-established cytosolic sensors and the corresponding innate immunity defenses beside the inflammasome and GBPs discussed here.
2. Figure 1: Why is the activation of caspase-1 impaired in *Gsdmd1105N/1105N* (Fig.1D, the last lane)? Does this mean caspase-1 cannot be activated if no GSDMD pore is formed on the cell membrane? Some explanations are needed, which could possibly make the story easier to follow. After all, ligands for the cytosolic innate immune sensors other than LOS derived from *M. catarrhalis* could be present and play some roles during infection. Similar circumstance occurs in Lines 160-161, the sentence "These results suggest that *M. catarrhalis* infection activates caspase-11 and this is followed by activation of the NLRP3-ASC inflammasome" need explanations in more details to fully support this conclusion.
3. Line 242, although the authors showed the evidence that OMV can deliver LOS into the host cytosol, LOS released into the cytosol by ways other than OMV could not be excluded and discussions are needed to clarify this point, as it is shown later that GBP2 causes the membrane rupture of *M. catarrhalis* and have bactericidal activity. I wonder what will happen in a cell infected with *M. catarrhalis* in term of LOS recognition by the host? Please describe the possible process of LOS recognition by caspase-11/4/5, which is facilitated by the interferon induced GBPs.
4. Figure 4: I am a little concerned about whether those *M. catarrhalis* bacteria shown in A and B are indeed intracellular as stated in the title or just attached to the cells? Although the correlative light electron microscopy images clearly demonstrate that GBP2 is colocalized with the intracellular bacteria. How do the authors ensure that they are analyzing the intracellular *M. catarrhalis* bacteria in experiments shown in Fig4 A and B? For the quantitation of GBPs-positive *M. catarrhalis* bacteria, information about the methods used and how many cells or bacteria are included in the analysis should be introduced in the legend.
5. Figure 7, Legend is missing group age and sex of mice, and number of times the experiment was repeated (should be repeated to show reproducibility). I am a little confused that it seems that bacterial loads in WT mice have been analyzed for many times (at least six times)? and the results from each time with 10 mice per group shows a significant variation ranging from less than 104 cfu to over 105 in bacterial load in spleen. Please provide some information in details about the animal infection experiments for the better understanding of data by the audience.

Minor points:

1. Line 169, the cited reference should not be in superscript.

2. Line 175, from my point of view, data with *Ninj1*^{-/-} BMDMs infected with *M. catarrhalis* seems to be not helpful for the conclusion drawn in this section.

3. Line 24, the sentence "Our transcriptomic and cytokine analyses revealed a type I IFN signature" need explanations in details about the type I IFN signature.

We are grateful to the reviewers for their constructive comments and valuable suggestions. We firmly believe that the reviewers' suggestions have substantially contributed to the overall quality of our Research Article. We hope that the revised manuscript is now suitable for publication.

Reviewer #1

The manuscript "Cytosolic immunity against *Moraxella catarrhalis* requires innate immune signalling and inflammasomes" by Tuipulotu et al. shows that GBPs compromises *M. catarrhalis* membrane integrity leading to the delivery of *M. catarrhalis* lipooligosaccharide (LOS) to the host cytoplasm by outer membrane vesicles (OMVs) which leads to activation of caspase-11 and subsequently to activation of NLRP3 and pyroptosis.

In the first part of the paper they show that CASP11 is upstream of the activation of NLRP3/CASP1, secretion of IL1b/IL18, and cell death. They go on to show that LOS and OMVs phenocopy the results obtained from infection with whole bacteria. They subsequently show that the inflammasome activation is IFNAR and GBPchr3 dependent, GBPs co-localize with the intracellular bacteria, and deletion of GBP2 has the largest effect on inflammasome activation. Recombinant GBP2 can kill bacteria and GBP2 appears to be able to disrupt the bacterial membrane. Mice lacking GBP1, 2, 3, 5, CASP11, or NLRP3 have higher bacterial burdens and lower levels of IL18 in the spleen.

This is a very detailed and quite complete manuscript that uncovers the mechanism by which *M. catarrhalis* induces the activation of innate immunity. Although the experiments were well performed and the conclusions are valid, the role of GBP2 in mediating the release of PAMPs from bacteria leading to CASP11 activation is not novel and has been described in several manuscripts.

Major comments:

1. It was unclear how the different GBPs that play a role (GBP1, 2, 3, 5) act together to activate the inflammasome. If GBP2 is sufficient to release PAMPs from the bacteria why does knocking out GBP 1, 3 or 5 also lead to a significant decrease in inflammasome activation?

We thank the reviewer for this comment. This is an important and difficult issue that will need to be addressed by substantial work as part of future studies investigating the roles of redundancies between GBPs during infection. We have recognised your comment and included additional discussion to further elaborate on these future directions (lines 386-400):

"The functional redundancies between GBPs in the activation of inflammasomes have been observed in macrophages infected with *Francisella novicida* (GBP1, GBP2, GBP3, GBP5 and GBP7), *Escherichia coli* (GBP2 and GBP5), and *Citrobacter rodentium* (GBP2 and GBP5) (Feng, Enosi Tuipulotu et al., 2022, Finethy, Luoma et al., 2017, Man, Karki et al., 2015, Man, Karki et al., 2016). In our study, the strongest reduction in inflammasome activation was observed in *Gbp2*^{-/-} BMDMs. However, *Gbp2*^{-/-} BMDMs did not have a complete loss of inflammasome responses, suggesting that other GBPs also in a nuanced way contribute to inflammasome activation in response to *M. catarrhalis*. Indeed, we observed that BMDMs lacking GBP1, GBP3 and GBP5 had a reduction in

inflammasome activation in response *M. catarrhalis* infection, highlighting that GBP2 alone is not solely responsible for inflammasome activation. A key question that has remained unanswered is how GBPs individually contribute to the host response. It is possible that each GBP is recruited to *M. catarrhalis* independently of one another, or that GBPs are recruited sequentially, with GBP2 as the apical GBP in the recruitment process. Following the recruitment of GBPs to the bacteria, GBPs may further stabilise one another to enable to the optimal release of PAMPs from bacteria and mediate the activation of inflammasomes.”

Minor comments:

2. Fig 1C the cytokine release is not indicated as fold-change over uninfected but as a concentration. It is unclear how this would confirm the gene expression data presented.

We thank the reviewer for this comment. The level of cytokines found in untreated controls are largely undetectable and therefore fold changes were not appropriate for this analysis. We have removed the word “confirm” when discussing the cytokine release data following the gene expression data.

Referee #2:

Tuipulotu et al. present a study characterising how the gram-negative human pathogen *Moraxella catarrhalis* activates cytosolic innate immune sensors. Despite being a significant human pathogen our understanding of how *M. catarrhalis* causes inflammation is relatively poor and thus this study addresses a clear knowledge gap. The authors convincingly demonstrate using genetic and pharmacological approaches that *M. catarrhalis* infection causes caspase-11 activation and thereby NLRP3 inflammasome activation in mouse macrophages and human THP-1 cells. The authors demonstrate that *M. catarrhalis* lipooligosaccharide (LOS) is necessary and sufficient to trigger caspase-11 activation and that LOS can be delivered into cells by outer membrane vesicles. They then shown that type I interferon signalling is necessary for caspase-11/NLRP3 activation which is associated with the enhanced expression of numerous GBPs in response to *M. catarrhalis* infection. Using a range of knockout mice, they show that GBP1, GBP2, GBP3, and GBP5 all contribute to the ability of *M. catarrhalis* to activate caspase-11/NLRP3. GBP1, GBP2, GBP3 and GBP5 are all observed to co-localise with *M. catarrhalis* but loss of GBP2 appears to have the most significant effect on caspase-11/NLRP3 activation. Mechanistically, GBP2 appears to be able to directly disrupt *M. catarrhalis* membranes to release LOS rather than directly interacting with LOS as has been shown for other GBPs with LPS. Finally, in vivo infection of mice with *M. catarrhalis* via intraperitoneal injection shows that deficiency of caspase-11, NLRP3, GBP1, GBP2, GBP3, or GBP5 all result in decreased inflammation and enhanced bacterial burden in the spleen. There is a huge amount of data included in this study and the experiments presented are very comprehensive and well controlled. However, I have a number of outstanding questions about the work that the authors may wish to address. Some minor comments are also outlined below.

Major questions:

1. Given that *M. catarrhalis* infects the respiratory tract and that GBPs associate with *M. catarrhalis* in infected lung epithelial cells - is caspase-11 activated by *M. catarrhalis* infection in lung epithelial cells?

We thank the reviewer for recognising the large amount of data in our manuscript and that our experiments are comprehensive and well controlled. Our current work focuses on macrophages and we feel that investigation into the biology of lung epithelial cells, which may have different inflammasome responses, would be more appropriate as part of another study. We have, however, outlined this important direction for our studies in the discussion (lines 360-363).

“Other important questions that remain unanswered from our work include whether caspase-5, GBPs and inflammasome signalling contribute to host defence against *M. catarrhalis* in multiple human cell types such as lung epithelial cells.”

2. Is caspase-4 activated in primary human monocytes/macrophages or in human epithelial cells?

We have shown that inflammasome signalling was induced in WT human THP1 cells in response to *M. catarrhalis* infection, whereas THP1 cells lacking caspase-4 have an impaired ability to secrete IL-1 β , IL-18 and LDH (Fig. EV2). These data suggest that caspase-4 is activated in human

macrophage-like cells. We feel that investigation into the biology of epithelial cells would be more appropriate as part of a future study.

3. What is the role of caspase-5 in *M. catarrhalis* infection in human cells?

We have tried to investigate this issue. Data from our caspase-5-KO THP-1 cells were inconsistent such that we were unable to draw robust conclusions and clarify the role of caspase-5 in human cells. We have discussed this limitation in our discussion (lines 360-363):

“Other important questions that remain unanswered from our work include whether caspase-5, GBPs and inflammasome signalling contribute to host defence against *M. catarrhalis* in multiple human cell types such as lung epithelial cells.”

4. Which GBPs are important during *M. catarrhalis* infection of human cells?

We and others have shown that human GBPs behave differently compared with their murine counterparts. For example, human GBP1 has been shown to bind directly with *Salmonella enterica* serovar Typhimurium and *Shigella flexneri* (Santos, Boucher et al., 2020, Wandel, Kim et al., 2020), however, in this study, we show that LOS is not required for the recruitment of mouse GBP1, GBP2, GBP3 and GBP5 to *M. catarrhalis*. Recombinant hGBP1 has been shown to lack bacteriolytic activity on its own (Gaudet, Zhu et al., 2021), whereas we have shown that recombinant mGBP1 can kill both *F. novicida* and *N. meningitidis* (Feng et al., 2022). These results may suggest that the biological activities of human and murine GBPs are different. Although we have identified the role of murine GBPs in the host response to *M. catarrhalis*, we expect that the biology of human GBPs could be very different, and this issue will be re-visited separately.

5. Is the IP infection of mice representative of a natural infection? Why was an inhaled or intratracheal infection route not performed?

M. catarrhalis is a human-adapted pathogen and studies on the pathogenesis and host response to *M. catarrhalis* infection have been limited by a lack of a reliable animal model that fully reflects human disease by this pathogen. For example, the tracheal infection model of *M. catarrhalis* has the limitation of inconsistent clearance of the bacteria and a failure to lead to pneumonia (Unhanand, Maciver et al., 1992). Our mouse model aims to investigate *Moraxella catarrhalis*-induced sepsis and systemic infection. We hope that the reviewer appreciates that it would take a significant amount of additional effort to develop a new mouse model of infection at this point of the study. We are hopeful that additional mouse models may be developed using other routes of infection as discussed (lines 402-405):

“Although we used an intraperitoneal injection to model the systemic spread of *M. catarrhalis*, the majority of *M. catarrhalis* infections occur within the respiratory tract. Therefore, inhalation and/or intratracheal infection routes could help elucidate the role of inflammasomes and GBPs in the lungs.”

Minor question/comments:

1. Line 79-80 "and that mice lacking TLRs can still mount an immune defence to control *M. catarrhalis* infection (Hassan et al., 2012)." In this study only TLR4-deficient mice were tested in vivo so this should be rephrased as "and that mice lacking TLR4 can still mount an immune defence to control *M. catarrhalis* infection (Hassan et al., 2012)."

We have followed the reviewer's suggestion and amended the text to:

"and that mice lacking TLR4 can still mount an immune defence to control *M. catarrhalis* infection"

2. Fig. 1C. Actual values for this data should be included as a supplemental table.

The transcript and cytokine data have been included in the Source Data.

3. Supp. Fig. 1 (and throughout the manuscript) please state how many technical replicates are performed in each independent biological replicate experiment.

We have updated the figure legends to state where technical replicates were performed for each biological replicate.

4. Fig. 1G The legend for these graphs is confusing - suggest rearranging so each graph has a legend.

Each graph in Fig 1G now has a legend.

5. Supp. Fig. 3F. The observation that NINJ1 deficiency does not affect LDH release is very intriguing, however the authors don't present any other cell death assays such as SYTOX staining. Could differences in LDH release be due to timing of these experiments? In S3F there is 40% cell death after 20 h infection when in e.g., Supp. Fig 3D there is 70/80% cell death after 10 h of infection?

Upon the suggestion of reviewer 3, we have now removed the *Ninj1*^{-/-} data from the manuscript.

6. Supp. Fig. 4A,B. Were there any significant differences between strains in cell death and cytokine release? Were these comparisons made?

We thank the reviewer for this comment. The cell death and cytokine release induced by different strains of *M. catarrhalis* are likely to be multifactorial, including the rate of bacterial uptake into BMDMs, the amount of OMVs being produced and the quantity of LOS shed by *M. catarrhalis*. These factors will lead to differences in inflammasome activation and cell death. Therefore, we were very mindful not to over-interpret our data and avoided drawing conclusions by comparing how different strains activated the inflammasome and cell death.

7. Supp. Fig. 4C,D. THP-1s are a monocytic cell line and if you PMA treat them that doesn't make them macrophages. At best they can be referred to as macrophage-like.

We have amended the text throughout the manuscript as suggested.

8. Supp. Fig. 4C,D. It would strengthen this human cell data if WT THP-1s infected with *M. catarrhalis* were examined with MCC950 and/or caspase inhibitors. Ideally, it would be good to repeat the OMV and IpxA mutant data in human cells.

We thank the reviewer for their suggestion. Although these experiments could reinforce the role of NLRP3 and caspase-1 in this study, we feel this experiment would be better suited for a separate and larger investigation into human inflammasome functions against *Moraxella catarrhalis* as discussed (lines 355-360).

“Although we provide genetic evidence that *Moraxella catarrhalis* activates caspase-4 in THP-1 cells, our findings could be further strengthened using an NLRP3 and/or caspase-1/11 inhibitor to further interrogate the inflammasome pathway activated by *M. catarrhalis* in human cells. Further, our study did not investigate whether *M. catarrhalis* OMVs or *M. catarrhalis* Δ IpxA can induce inflammasome activation in THP-1 cells, and therefore further work is required to characterise the inflammasome signalling pathway in human cells.”

9. I don't think it's necessary to include Supp Fig 5J.

We have removed this figure from the manuscript.

10. How were the LA-4 cells that express tagged-GBP generated? There are no details on this in the methods.

We have now included the methodology for the generation of LA-4 cells in the methods section (lines 629-641).

“Generation of LA-4 cell lines with doxycycline-inducible expression of GBPs

To produce lentiviruses for the generation of cell lines with doxycycline (DOX) inducible expression of GBPs, a two-step cloning strategy was used. mGBP1 (Accession number NM_010259.2) was engineered to express an N-terminal FLAG-tag and cloned into the pTRE-tight plasmid vector, and then sub-cloned into the pFUV1-mCherry lentivirus transfer plasmid (Herold, van den Brandt et al., 2008) (kindly provided by Prof Marco Herold, The Walter and Eliza Hall Institute of Medical Research, Parkville, Australia). A cell line expressing cytoplasmic hen egg ovalbumin (cOVA) lacking the sequence for cell surface trafficking was prepared as a control. Lentivirus stocks to be used for subsequent LA-4 cell transduction were generated in 293T cells following transfection of (i) pMDL (gag and pol), (ii) pRSV-REV packaging plasmids, (iii) pMD2G.VSVg envelope plasmid and (iv) pFUV1-mCherry transfer plasmid by standard procedures. Lentivirus transduced LA-4 cells were sorted based on mCherry-positive cells using a FACSAria III instrument (BD Biosciences, New Jersey, USA).”

11. Supp Fig 8D - How is this quantification performed and how is a single bacterium defined? The quantification doesn't look consistent with the staining shown in S8C e.g., in the GBP-1 representative image it looks like approx. 50% the bacteria are GBP positive but this is 80-100% in the graph.

To quantify the bacterial number in a field of view, we use the single green channel (anti-*M. catarrhalis* staining) where a single bacterium is defined as an intact circle. To quantify the proportion

of GBP-positive bacteria in a field of view, we use the single red channel (anti-GBP staining) and compare this to the green channel (anti-*M. catarrhalis* staining) rather than using the merged image for quantitation which, depending on the strength of the fluorescence signal, can mask some bacteria positively-stained for GBPs resulting in inaccurate quantitation. The number of *M. catarrhalis* that are GBP-positive is then divided by the total number of *M. catarrhalis* counted to obtain the proportion of GBP-positive bacteria. The quantitation of GBP-positive bacteria in Supp Fig. 8D (now Fig. EV4) is therefore a representation of the GBP staining observed in the red channel (anti-GBP). We thank the reviewer for raising this point and have updated the figure legend to describe how our quantification was performed (lines 1369-1380).

“Data information: Each symbol represents an independent biological replicate (B). NS, no statistical significance (one-way ANOVA with Dunnett’s multiple-comparisons test (B and D)). To quantify the bacterial number in a field of view, we used the single green channel (anti-*M. catarrhalis* staining), where a single bacterium is defined as an intact circle. To quantify the proportion of GBP-positive bacteria in a field of view, we used the single red channel (anti-GBP staining) and compared this to the green channel (anti-*M. catarrhalis* staining). The number of GBP-positive *M. catarrhalis* was divided by the total number of *M. catarrhalis* counted to obtain the proportion of GBP-positive bacteria. A total of 100 lung epithelial cells (LA-4) were analysed to quantify the proportion of GBP-positive bacteria per cell using confocal microscopy (D). Data are from one experiment representative of two (C) or three independent experiments (A) or are pooled from two (D) or three independent experiments (B; mean and s.e.m. in B and D). Scale bars, 4 μ m (C).”

12. Fig. 4F and G. The different cell types used here should be labelled and the interpretation of this data is confusing. In Fig 4G most of the cells should be dead in the WT BMDM so it's unsurprising that there are more intracellular bacteria in the KOs which don't die.

We have included additional cell-type labels for both panels as suggested by the reviewer.

13. Supp. Fig. 10G. Additional data from this experiment should be shown (cell death, IL-1b)

We have now included data for IL-1 β and LDH in this figure as suggested by the reviewer.

14. "type I IFN signal driving activation of the caspase-11-NLRP3 inflammasome during *M. catarrhalis* infection." This sentence doesn't make sense. How are GBPs driving type1 IFN signalling?

We have revised this sentence for improved clarity (lines 313-315):

“These findings suggest that GBP2, via its membrane-disruptive activity instead of LOS binding, disrupts the membrane of *M. catarrhalis* to release LOS and facilitate activation of the caspase-11-NLRP3 inflammasome.”

Referee #3

Previous studies have shown that caspase-11 in mice and caspase-4/5 in humans are cytosolic innate immune sensors for LPS and recognition of LPS by these sensors leads to the noncanonical inflammasome activation and pyroptosis executed by GSDMD. Pore formation mediated by N-terminal domain of GSDMD allows not only the release of cytokines of the IL-1 family but also endogenous DAMPs from pyroptotic cells. In addition, interferon induced guanylate-binding proteins (GBPs) have been shown to recognize pathogen-containing vacuoles (*Toxoplasma gondii*, *Listeria monocytogenes*, etc.) or bind directly to the surface of pathogen (*Shigella flexneri*) residing in the host cytosol, leading to lysis of this intracellular niche, disruption of bacterial membrane integrity and induction of inflammasome. However, little has been known for the roles of inflammasome and GBPs in host defense against *Moraxella catarrhalis*, a major causative agent of otitis media and chronic obstructive pulmonary disease. Here, Tuipulotu et al present data suggesting that outer membrane vesicles introduced lipooligosaccharide of *M. catarrhalis* into the host cytosol activates the caspase-4/11, gasdermin-D-dependent pyroptosis, followed by the NLRP3 inflammasome activation in both human and mouse macrophages. They further showed the evidences that GBPs, especially GBP2 in mice, are critical for the inflammasome activations in host cells post *M. catarrhalis* infection and GBP2 can lead to the rupture of *M. catarrhalis* cell membrane and even shows killing activity to this bacteria. Finally, using the animal infection experiments, the authors showed the essentiality of inflammasomes and GBPs for the host defense against *M. catarrhalis*. This study overall is well designed and reports previously unrevealed roles of canonical and non-canonical inflammasome as well GBPs in fighting against *M. catarrhalis*, which will provide valuable information on *M. catarrhalis* and the LOS/LPS activated non-canonical inflammation, as well as the roles played by GBPs. There are still some concerns about this study as listed below.

Major points:

1. The title of the manuscript should be more specific and focused on the cytosolic innate immunity against *M. catarrhalis* discussed in this study, since there are a lots of well-established cytosolic sensors and the corresponding innate immunity defences beside the inflammasome and GBPs discussed here.

The title of the manuscript has been revised as suggested by the reviewer (lines 1-2):

“Cytosolic innate immunity against *Moraxella catarrhalis* requires guanylate-binding proteins and inflammasome activation.”

2. Figure 1: Why is the activation of caspase-1 impaired in Gsdmdl105N/I105N (Fig.1D, the last lane)? Does this mean caspase-1 cannot be activated if no GSDMD pore is formed on the cell membrane? Some explanations are needed, which could possibly make the story easier to follow. After all, ligands for the cytosolic innate immune sensors other than LOS derived from *M. catarrhalis* could be present and play some roles during infection. Similar circumstance occurs in Lines 160-161, the sentence "These results suggest that *M. catarrhalis* infection activates caspase-11 and this is followed by activation of the NLRP3-ASC inflammasome" need explanations in more details to fully support this conclusion.

Gsdmd^{105N/105N} mice carry a loss-of-function mutation in gasdermin D, which does not impair proteolytic cleavage of GSDMD but prevents pore formation in the plasma membrane (Kagayaki et al. 2011). This information is mentioned in both the methods and results text and helps explain why activation of caspase-1 is impaired in *Gsdmd*^{105N/105N} BMDMs. *M. catarrhalis* activates caspase-11, which cleaves *Gsdmd* into a non-functional p30 fragment that cannot induce activation of the downstream pathway.

3. Line 242, although the authors showed the evidence that OMV can deliver LOS into the host cytosol, LOS released into the cytosol by ways other than OMV could not be excluded and discussions are needed to clarify this point, as it is shown later that GBP2 causes the membrane rupture of *M. catarrhalis* and have bactericidal activity. I wonder what will happen in a cell infected with *M. catarrhalis* in term of LOS recognition by the host? Please describe the possible process of LOS recognition by caspase-11/4/5, which is facilitated by the interferon induced GBPs.

We thank the reviewer for giving us the opportunity to update our discussion to highlight the potential ways in which LOS from *M. catarrhalis* can be delivered and recognised by the cells (lines 346-353).

“Our data provides evidence that OMVs can deliver LOS to the host cytoplasm to induce activation of the caspase-11-NLRP3 inflammasome. However, several mechanisms by which LOS is delivered to the host cytoplasm may exist. For example, internalised *M. catarrhalis* likely delivers LOS to the host cytoplasm and extracellular LOS from *M. catarrhalis* may be internalised by CD14 and HMGB1 into the cytoplasm independently of TLR4 internalisation (Deng, Tang et al., 2018, Vasudevan, Russo et al., 2022). Given that the lipid A portion of cytoplasmic LPS is recognised by caspase-4/5/11 (Hagar, Powell et al., 2013, Shi, Zhao et al., 2014), it is likely that lipid A from LOS is also recognised in a similar manner.”

4. Figure 4: I am a little concerned about whether those *M. catarrhalis* bacteria shown in A and B are indeed intracellular as stated in the title or just attached to the cells? Although the correlative light electron microscopy images clearly demonstrate that GBP2 is colocalized with the intracellular bacteria. How do the authors ensure that they are analysing the intracellular *M. catarrhalis* bacteria in experiments shown in Fig. 4A and B? For the quantitation of GBPs-positive *M. catarrhalis* bacteria, information about the methods used and how many cells or bacteria are included in the analysis should be introduced in the legend.

Extracellular bacteria and bacteria attached to the cell surface were removed by washing with PBS three times and incubated in gentamycin-containing media (50 µg/mL) for 1 hour, followed by washing three times with PBS to ensure removal of extracellular and attached bacteria before immunofluorescence staining and confocal microscopy.

In Fig. 4B, each dot represents the proportion of GBP-positive bacteria quantified in a single BMDM. For quantitation of GBP-positive *M. catarrhalis*, 100 BMDMs were analysed. To quantify the bacterial number in a field of view, we use the single green channel (anti-*M. catarrhalis* staining) where a single bacterium is defined as an intact circle. To quantify the proportion of GBP-positive bacteria in a field of view, we use the single red channel (anti-GBP staining) and compare this to the green channel (anti-

M. catarrhalis staining), rather using the merged image for quantitation which, depending in the strength of the fluorescence signal, can mask some bacteria positively-stained for GBPs resulting in inaccurate quantitation. The number of *M. catarrhalis* that are GBP-positive was divided by the total number of *M. catarrhalis* counted to obtain the proportion of GBP-positive bacteria. We have updated the information in the figure legend to describe how our quantification was performed (lines 1225-1237):

“Data information: Each symbol represents an independent biological replicate (E and F). NS, no statistical significance; * $P < 0.05$; ** $P < 0.01$; *** $P < 0.001$; **** $P < 0.0001$ (one-way ANOVA with Dunnett’s multiple-comparisons test (E-G). To quantify the bacterial number in a field of view, we used the single green channel (anti-*M. catarrhalis* staining), where a single bacterium is defined as an intact circle. To quantify the proportion of GBP-positive bacteria in a field of view, we used the single red channel (anti-GBP staining) and compared this to the green channel (anti-*M. catarrhalis* staining). The number of GBP-positive *M. catarrhalis* was divided by the total number of *M. catarrhalis* counted to obtain the proportion of GBP-positive bacteria (B). A total of 100 BMDMs were analysed to quantify the proportion of GBP-positive bacteria per cell using confocal microscopy (B) or the intracellular bacterial burden using electron microscopy (G). Data are from one experiment representative of two (G) or three independent experiments (A, C, D) or are pooled from two (B) or three independent experiments (E, F; mean and s.e.m. in B, E-G). Scale bars, 5 μm (A, C).”

5. Figure 7, Legend is missing group age and sex of mice, and number of times the experiment was repeated (should be repeated to show reproducibility). I am a little confused that it seems that bacterial loads in WT mice have been analysed for many times (at least six times)? and the results from each time with 10 mice per group shows a significant variation ranging from less than 10^4 CFU to over 10^5 in bacterial load in spleen. Please provide some information in details about the animal infection experiments for the better understanding of data by the audience.

We have followed the reviewer’s advice and updated the figure legend to include age and sex of the mice and the number of times each experiment was independently repeated (lines 1296-1300). As pointed out by the reviewer, the inherent level of biological variation in bacterial burden between experiments exist such that we were not able to pool data from multiple experiments. We were also concerned about the large number of mice used for animal ethics and welfare reasons, and therefore, used mice from both sexes and within the age range of 6 to 8 weeks to minimise the number of and welfare impact on animals.

“Data information: A mix of male and female mice 6-8 weeks old were used in each experiment and were injected i.p. with 2×10^7 CFU of *M. catarrhalis* and analysed after 6 h. Each symbol represents an individual mouse (A-L). Each panel represents data from a single experiment. Each experiment was performed at least two times. * $P < 0.05$; ** $P < 0.01$; *** $P < 0.001$; **** $P < 0.0001$ (two-tailed t -test (A-L)).”

Minor points:

6. Line 169, the cited reference should not be in superscript.

This has been fixed.

7. Line 175, from my point of view, data with *Ninj1*^{-/-} BMDMs infected with *M. catarrhalis* seems to be not helpful for the conclusion drawn in this section.

We have now removed *Ninj1*^{-/-} data from the manuscript.

8. Line 224, the sentence "Our transcriptomic and cytokine analyses revealed a type I IFN signature" need explanations in details about the type I IFN signature.

We have revised this sentence as suggested by the reviewer (lines 233-235).

"Our transcriptomic and cytokine analyses revealed a type I IFN signature characterised by the upregulation of interferon-stimulated genes, such as *Nos2*, *Rsad2*, *Il33*, *Lif*, *Isg15*, *Trim30c*, *Ifit1*, *Oas2*."

References

- Correa-Martinez CL, Rauwolf KK, Schuler F, Fuller M, Kampmeier S, Groll AH (2019) *Moraxella nonliquefaciens* bloodstream infection and sepsis in a pediatric cancer patient: case report and literature review. *BMC Infect Dis* 19: 836
- Deng M, Tang Y, Li W, Wang X, Zhang R, Zhang X, Zhao X, Liu J, Tang C, Liu Z, Huang Y, Peng H, Xiao L, Tang D, Scott MJ, Wang Q, Liu J, Xiao X, Watkins S, Li J et al. (2018) The Endotoxin Delivery Protein HMGB1 Mediates Caspase-11-Dependent Lethality in Sepsis. *Immunity* 49: 740-753 e7
- Feng S, Enosi Tuipulotu D, Pandey A, Jing W, Shen C, Ngo C, Tessema MB, Li FJ, Fox D, Mathur A, Zhao A, Wang R, Pfeffer K, Degrandi D, Yamamoto M, Reading PC, Burgio G, Man SM (2022) Pathogen-selective killing by guanylate-binding proteins as a molecular mechanism leading to inflammasome signaling. *Nat Commun* 13: 4395
- Finethy R, Luoma S, Orench-Rivera N, Feeley EM, Haldar AK, Yamamoto M, Kanneganti TD, Kuehn MJ, Coers J (2017) Inflammasome Activation by Bacterial Outer Membrane Vesicles Requires Guanylate Binding Proteins. *mBio* 8
- Funaki T, Inoue E, Miyairi I (2016) Clinical characteristics of the patients with bacteremia due to *Moraxella catarrhalis* in children: a case-control study. *BMC Infect Dis* 16: 73
- Gaudet RG, Zhu S, Halder A, Kim BH, Bradfield CJ, Huang S, Xu D, Maminska A, Nguyen TN, Lazarou M, Karatekin E, Gupta K, MacMicking JD (2021) A human apolipoprotein L with detergent-like activity kills intracellular pathogens. *Science* 373
- Gomis RM, Fos, II, Gomis CV, Rubio J (2010) [Moraxella catarrhalis sepsis in a 4 month-old infant with RSV+ bronchiolitis]. *An Pediatr (Barc)* 72: 151
- Hagar JA, Powell DA, Aachoui Y, Ernst RK, Miao EA (2013) Cytoplasmic LPS activates caspase-11: implications in TLR4-independent endotoxic shock. *Science* 341: 1250-3
- Herold MJ, van den Brandt J, Seibler J, Reichardt HM (2008) Inducible and reversible gene silencing by stable integration of an shRNA-encoding lentivirus in transgenic rats. *Proc Natl Acad Sci U S A* 105: 18507-12
- Ioannidis JP, Worthington M, Griffiths JK, Snyderman DR (1995) Spectrum and significance of bacteremia due to *Moraxella catarrhalis*. *Clin Infect Dis* 21: 390-7
- Man SM, Karki R, Malireddi RK, Neale G, Vogel P, Yamamoto M, Lamkanfi M, Kanneganti TD (2015) The transcription factor IRF1 and guanylate-binding proteins target activation of the AIM2 inflammasome by *Francisella* infection. *Nat Immunol* 16: 467-75
- Man SM, Karki R, Sasai M, Place DE, Kesavardhana S, Temirov J, Frase S, Zhu Q, Malireddi RKS, Kuriakose T, Peters JL, Neale G, Brown SA, Yamamoto M, Kanneganti TD (2016) IRGB10 Liberates Bacterial Ligands for Sensing by the AIM2 and Caspase-11-NLRP3 Inflammasomes. *Cell* 167: 382-396 e17
- Nakayama A, Yamanaka K, Hayashi H, Ohkusu K (2014) *Moraxella lacunata* infection associated with septicemia, endocarditis, and bilateral septic arthritis in a patient undergoing hemodialysis: a case report and review of the literature. *J Infect Chemother* 20: 61-4
- Santos JC, Boucher D, Schneider LK, Demarco B, Dilucca M, Shkarina K, Heilig R, Chen KW, Lim RYH, Broz P (2020) Human GBP1 binds LPS to initiate assembly of a caspase-4 activating platform on cytosolic bacteria. *Nat Commun* 11: 3276
- Shah SS, Ruth A, Coffin SE (2000) Infection due to *Moraxella osloensis*: case report and review of the literature. *Clin Infect Dis* 30: 179-81
- Shi J, Zhao Y, Wang Y, Gao W, Ding J, Li P, Hu L, Shao F (2014) Inflammatory caspases are innate immune receptors for intracellular LPS. *Nature* 514: 187-92
- Thorsson B, Haraldsdottir V, Kristjansson M (1998) *Moraxella catarrhalis* bacteraemia. A report on 3 cases and a review of the literature. *Scand J Infect Dis* 30: 105-9
- Unhanand M, Maciver I, Ramilo O, Arencibia-Mireles O, Argyle JC, McCracken GH, Jr., Hansen EJ (1992) Pulmonary clearance of *Moraxella catarrhalis* in an animal model. *J Infect Dis* 165: 644-50
- Vasudevan SO, Russo AJ, Kumari P, Vanaja SK, Rathinam VA (2022) A TLR4-independent critical role for CD14 in intracellular LPS sensing. *Cell Rep* 39: 110755

Wandel MP, Kim BH, Park ES, Boyle KB, Nayak K, Lagrange B, Herod A, Henry T, Zilbauer M, Rohde J, MacMicking JD, Randow F (2020) Guanylate-binding proteins convert cytosolic bacteria into caspase-4 signaling platforms. *Nat Immunol* 21: 880-891

Westendorp IC, Tiemessen MA, de Jong M, Soomers A, Wakelkamp IM, Boersma WG (2005) *Moraxella catarrhalis* sepsis in a patient with juvenile spinal muscle atrophy. *Neth J Med* 63: 227-9

Woodbury A, Jorgensen J, Owens A, Henao-Martinez A (2009) *Moraxella lacunata* septic arthritis in a patient with lupus nephritis. *J Clin Microbiol* 47: 3787-8

Dear Si Ming,

Thank you for submitting your revised manuscript to The EMBO Journal. Your study has now been re-reviewed by referees #2 and 3. As you can see from the comments below, both referees appreciate the introduced changes and support publication here. They have a few remaining points that can be addressed with text changes. When you return the revised version will you also resolve the following editorial points:

- Reference list - for articles with more than 10 authors please cut after 10 authors followed by et al.
- Please remove the Authors Contributions from the manuscript. The 'Author Contributions' section is replaced by the CRediT contributor roles taxonomy to specify the contributions of each author in the journal submission system. Please use the free text box in the 'author information' section of the manuscript submission system to provide more detailed descriptions (e.g., 'X provided intracellular Ca⁺⁺ measurements in fig Y')
- Please check figure callout for Figure 7J
- Our publisher has also done their pre-publication check on your manuscript. When you log into the manuscript submission system you will see the file "Data Edited Manuscript file". Please take a look at the word file and the comments regarding the figure legends and respond to the issues.
- We don't encourage showing statistic when n=2 (Fig 1G). This is also related to the remaining comment raised by referee #2.

Please also submit a point-by-point response when you send in your revised version.

Congratulations on a nice study - almost there!!

Best Karin

Karin Dumstrei, PhD
Senior Editor
The EMBO Journal

Guide For Authors: <https://www.embopress.org/page/journal/14602075/authorguide>

Use the link below to submit your revision:

Referee #2:

In their rebuttal the authors have argued that answering any of my major questions is outside the scope of the current work. I certainly appreciate this point and don't want to any unnecessary experiments to be performed. However, I feel that the lack of human cell data and experiments with other cell types that are relevant to infection limits the impact of this study on an important human pathogen. This is nevertheless a comprehensive piece of work on this *Moraxella catarrhalis* in a mouse system and it will certainly inspire further work on this pathogen.

I have one outstanding issue relating to my minor comment 3 on the number of technical replicates that were performed. The authors have only added technical replicate details in Figure 6J. Does this mean that for example in Figure 1E each symbol is from one well of a TC plate and that no technical replicates were performed within each independent experiment? Were cells from one mouse stimulated in one well of a TC plate and then this supernatant was analysed by ELISA? Was this then repeated on three separate occasions? The authors need to clarify exactly how these experiments were conducted as this is unclear from the methods and figure legends.

Referee #3:

The authors have answered all the points raised and added additional clarification to the revised manuscript. However, I think

there are still some issues with revised manuscript that need to be resolved.

1. All the figures have not been numbered and it is a little confusing to find the results corresponding to the descriptions in the manuscript.
2. Figure 4E, it is not clear why GBPchr3-KO BMDMs transfected with LPS secrete IL-1 β at a significantly lower level? The authors stated that GBPs might induce bacterial rupture to facilitate LOS release to the host cytosol, could the authors make some explanations on what roles are GBPs playing under this condition, especially on those other than facilitating LOS recognition in inflammasome activation.
3. Abstract, I suggest the authors amending the description of LOS delivery into the host cytosol by OMV. Based on the infection experiments of GBPs-KO mice shown in Figure 7, I prefer that mGBP2, along with other mGBPs, might play more important role than OMV in facilitating LOS recognition and Casp-11-NLRC3 inflammasome signaling during host infection. The statement "We show that *M. catarrhalis* outer membrane vesicles introduce lipooligosaccharide (LOS) into the host cell cytoplasm to activate the cytosolic innate immune sensor caspase-4/11, gasdermin-D-dependent pyroptosis, and the NLRP3 inflammasome in human and mouse macrophages" could mislead the audience to deem that OMV delivery is the major way of LOS to enter into the host cytosol. Actually, it is likely that OMV merely play a relatively minor role.

We are grateful to the reviewers for their constructive comments and valuable suggestions. We firmly believe that the reviewers' suggestions have substantially contributed to the overall quality of our Research Article. We hope that the revised manuscript is now suitable for publication.

Reviewer #2

In their rebuttal the authors have argued that answering any of my major questions is outside the scope of the current work. I certainly appreciate this point and don't want to any unnecessary experiments to be performed. However, I feel that the lack of human cell data and experiments with other cell types that are relevant to infection limits the impact of this study on an important human pathogen. This is nevertheless a comprehensive piece of work on this *Moraxella catarrhalis* in a mouse system and it will certainly inspire further work on this pathogen.

1. I have one outstanding issue relating to my minor comment on the number of technical replicates that were performed. The authors have only added technical replicate details in Figure 6. Does this mean that for example in Figure 1E each symbol is from one well of a TC plate and that no technical replicates were performed within each independent experiment? Were cells from one mouse stimulated in one well of a TC plate and then this supernatant was analysed by ELISA? Was this then repeated on three separate occasions? The authors need to clarify exactly how these experiments were conducted as this is unclear from the methods and figure legends.

Each data point in Figure 1E represents an independent biological replicate, where no technical repeats were performed. As suggested by the reviewer, we have now amended the methods section to provide further information on how our experiments were conducted (Under the new heading "Data collection and statistical analysis").

"At least three independent biological repeats were performed for each experiment unless otherwise stated in the figure legend. Experiments were performed without technical replicates unless otherwise stated in the figure legend. For example, cells from one mouse were stimulated in one well of a tissue-culture plate and analysed using various techniques. This was then repeated on at least three separate occasions unless otherwise stated in the figure legend."

Referee #3

The authors have answered all the points raised and added additional clarification to the revised manuscript. However, I think there are still some issues with revised manuscript that need to be resolved.

1. All the figures have not been numbered and it is a little confusing to find the results corresponding to the descriptions in the manuscript.

All figures have now been numbered.

2. Figure 4E, it is not clear why GBPchr3-KO BMDMs transfected with LPS secrete IL-1 β at a significantly lower level? The authors stated that GBPs might induce bacterial rupture to facilitate LOS release to the host cytosol, could the authors make some explanations on what roles are GBPs playing under this condition, especially on those other than facilitating LOS recognition in inflammasome activation.

We thank the reviewer for these questions. Previous studies have shown that IL-1 β secretion and LDH release are decreased in *Gbp^{chr3}*-KO BMDMs after LPS transfection at early time points, but this dependency on GBPs is reduced or abolished at later time points [1-4]. Here, we observed a significant reduction in IL-1 β secretion in *Gbp^{chr3}*-KO BMDMs following LPS transfection for 5 hours (Fig. 4E), however, this phenotype was abolished when LPS transfection was performed for 10 hours (Fig. EV4). These findings are consistent with previous studies that have shown a loss of GBP dependency for non-canonical inflammasome activation following LPS transfection over time [2-4]. However, exactly why we see a reduction specifically in the secretion of IL-1 β is unclear. It is possible that endogenous GBPs might rupture liposome-packaged LPS to allow enhanced LPS dissemination within the cytoplasm.

With regards to the reviewer's question about the roles of GBPs other than facilitating LOS recognition and inflammasome activation, we have discussed this further in the manuscript (lines 385-390).

"Our study suggests that GBPs induce bacterial rupture to facilitate LOS release for release into the cytoplasm for activation of the inflammasome, however it is likely that GBP-mediated rupture may also release other bacterial ligands apart from LOS for detection by cytosolic immune sensors. Given our finding that cGAS and STING partly contribute to IFN- β production in response to *M. catarrhalis* infection, it is possible that GBPs mediate the release of DNA from *M. catarrhalis* for detection by these sensors to amplify IFN-GBP-inflammasome signalling."

3. Abstract, I suggest the authors amending the description of LOS delivery into the host cytosol by OMV. Based on the infection experiments of GBPs-KO mice shown in Figure 7, I prefer that mGBP2, along with other mGBPs, might play more important role than OMV in facilitating LOS recognition and Casp-11-NLRC3 inflammasome signaling during host infection. The statement "We show that *M. catarrhalis* outer membrane vesicles introduce lipooligosaccharide (LOS) into the host cell cytoplasm to activate the cytosolic innate immune sensor caspase-4/11, gasdermin-D-dependent pyroptosis, and the NLRP3 inflammasome in human and mouse macrophages" could mislead the audience to deem that OMV

delivery is the major way of LOS to enter into the host cytosol. Actually, it is likely that OMV merely play a relatively minor role.

We have removed the mention of LOS delivery by OMVs from the abstract as suggested by the reviewer. We also added the mention of GBP2 (lines 45-50).

“We show that *M. catarrhalis* and its outer membrane vesicles or lipooligosaccharide (LOS) can activate the cytosolic innate immune sensor caspase-4/11, gasdermin-D-dependent pyroptosis, and the NLRP3 inflammasome in human and mouse macrophages.... We also show that inflammasomes and GBPs, particularly GBP2, are required for the host defence against *M. catarrhalis* in mice.”

References

1. Brubaker, S.W., et al., *A Rapid Caspase-11 Response Induced by IFN γ Priming Is Independent of Guanylate Binding Proteins*. iScience, 2020. **23**(10): p. 101612.
2. Meunier, E., et al., *Caspase-11 activation requires lysis of pathogen-containing vacuoles by IFN-induced GTPases*. Nature, 2014. **509**(7500): p. 366-70.
3. Pilla, D.M., et al., *Guanylate binding proteins promote caspase-11-dependent pyroptosis in response to cytoplasmic LPS*. Proc Natl Acad Sci U S A, 2014. **111**(16): p. 6046-51.
4. Santos, J.C., et al., *LPS targets host guanylate-binding proteins to the bacterial outer membrane for non-canonical inflammasome activation*. EMBO J, 2018. **37**(6).

Dear Si Ming,

Thank you for submitting your revised manuscript to The EMBO Journal. I have now had a chance to take a careful look at everything and all looks good. I am therefore very pleased to accept the manuscript for publication here.

Congratulations on a nice study!

Best Karin

Karin Dumstrei, PhD
Senior Editor
The EMBO Journal

Please note that it is EMBO Journal policy for the transcript of the editorial process (containing referee reports and your response letter) to be published as an online supplement to each paper. If you do NOT want this, you will need to inform the Editorial Office via email immediately. More information is available here:

<https://www.embopress.org/page/journal/14602075/authorguide#transparentprocess>

Your manuscript will be processed for publication in the journal by EMBO Press. Manuscripts in the PDF and electronic editions of The EMBO Journal will be copy edited, and you will be provided with page proofs prior to publication. Please note that supplementary information is not included in the proofs.

You will be contacted by Wiley Author Services to complete licensing and payment information. The required 'Page Charges Authorization Form' is available here: https://www.embopress.org/pb-assets/embo-site/tej_apc.pdf - please download and complete the form and return to embopressproduction@wiley.com

EMBO Press participates in many Publish and Read agreements that allow authors to publish Open Access with reduced/no publication charges. Check your eligibility: <https://authorservices.wiley.com/author-resources/Journal-Authors/open-access/affiliation-policies-payments/index.html>

Should you be planning a Press Release on your article, please get in contact with embojournal@wiley.com as early as possible, in order to coordinate publication and release dates.

If you have any questions, please do not hesitate to call or email the Editorial Office. Thank you for your contribution to The EMBO Journal.
